# Intraspecific genetic variability and diurnal activity affect environmental DNA detection in Japanese eel

Sayaka Takahashi[1,2,3]*, Shingo Takada[4], Hiroki Yamanaka[5,6], Reiji Masuda[3], Akihide Kasai[7]

1 Oita Junior College, Oita, Japan, 2 Faculty of Life and Environmental Science, Shimane University, Matsue, Shimane, Japan, 3 Maizuru Fisheries Research Station, Kyoto University, Nagahama, Maizuru, Kyoto, Japan, 4 Division of Marine Bioresource and Environmental Science, Graduate School of Fisheries Sciences, Hokkaido University, Hakodate, Hokkaido, Japan, 5 Faculty of Advanced Science and Technology, Ecology and Environmental Engineering Course, Ryukoku University, Otsu, Shiga, Japan, 6 Center for Biodiversity Science, Ryukoku University, Otsu, Shiga, Japan, 7 Faculty of Fisheries Sciences, Division of Marine Bioresource and Environmental Science, Hokkaido University, Hakodate, Hokkaido, Japan

* takahashi.sayaka.k35@kyoto-u.jp

**Data Availability Statement:** All relevant data are within the manuscript and its Supporting information files.

**Funding:** This work was supported by the CREST program of the Japan Science and Technology

## Abstract

Environmental DNA (eDNA) analysis with species-specific primer/probe sets is promising as a tool to quantify fish abundance and distribution. Nevertheless, several factors could reduce the accuracy of this method. Here, we aimed to analyze whether intraspecific variability and diel activity rhythm affect eDNA detection in Japanese eels (*Anguilla japonica*). For this purpose, we performed tank experiments focusing on two points. First, we assessed the effects of base pair sequences with probe region polymorphism on eDNA detection. Next, we evaluated the influences of diel rhythm, activity, and individual differences in eDNA release rate on eDNA concentration. We examined the base pair sequences of the probe regions of 20 individuals and found genetic mismatches in two of them. The eDNA concentration was estimated to be much lower in these variants than it was in the other individuals. We conducted a rearing experiment on four non-variant individuals to explore the influences of diel activity and inter-individual differences in eDNA detection. Nocturnal eel activity was reflected in the eDNA detection but the inter-individual differences remained large. The observed weak positive correlations between eDNA concentration and activity suggest that eDNA emission is highly dependent on basal metabolism. The present study suggests that consideration of polymorphic sites at the probe region and diel activity rhythms should improve the accuracy and precision of abundance estimation through eDNA. Such fine-tuning is applicable not only for eels but also for other fishes to be targeted by eDNA technology.

## Introduction

Since antiquity, Japanese eel (*Anguilla japonica*) has been an important food fish in East Asian countries. However, its populations have dramatically declined in recent decades because of

Agency (Grant No. JPMJCR13A2), JSPS KAKENHI (Grant No. 17H01412), and JSPS KAKENHI (Grant No. 19H05641). The funders had no role in study design, data collection and analysis, decision to publish, or preparation of the manuscript.

**Competing interests:** The authors have declared that no competing interests exist.

climate change and overexploitation [1]. Consequently, it has been listed as "an endangered species in the near future" [2]. To protect natural Japanese eel populations, it is necessary to determine when, where, and how many animals are located in each habitat. Japanese eels spawn near the Mariana Islands [3]. The newly hatched preleptocephali and more developed leptocephali are transported northward by the Kuroshio Current [4]. They then metamorphose into glass eels when they reach rivers or brackish water, develop into elvers, become yellow and silver eels, and return to the offshore spawning site [5]. However, the precise spatiotemporal distributions of the eels in various river sites are yet to be established. Today, environmental DNA (eDNA) analyses have been applied in the establishment of eel distribution [6,7].

Analyses of eDNA estimate target species composition and abundance using DNA fragments left by animals and plants in water, soil, and air [8–12]. The technique has been used for spatiotemporal monitoring of endangered land and aquatic animal species [13] such as marine sturgeon [14] and freshwater carp [15]. It has also been applied to detect the reproductive activity of endangered species such as freshwater Macquarie perch [16]. It is necessary to identify factors that influence eDNA detection when eDNA analysis is used to quantify species.

In quantitative PCR (qPCR) analyses using species-specific primers/probes, mutations and polymorphisms may influence eDNA detection rate. Base pair mismatches between target species and probes have been reported to reduce the detection of DNA in humans [17] or primers in fish [18]. In addition, haplotype diversity in Japanese eel [19] may affect the sensitivity of eDNA detection activities. This is a fundamental problem that should be adequately addressed when developing eDNA assays.

Conversely, ecological and physiological traits of target species could pose challenges when eDNA concentrations are used as proxies in biomass estimation activities. Some field studies demonstrated that there is a positive correlation between eDNA concentration and fish biomass/abundance in freshwater [13,20–24] and marine [25–28] environments. Tank experiments have also demonstrated positive correlations between fish size and eDNA concentrations in bluegill [29] and fish abundance and eDNA concentrations in jack mackerel [30]. Nevertheless, other researchers reported only weak quantitative relationships between biomass and eDNA abundance [6,31–33]. Individual differences, such as metabolic rate, stress conditions, life stage, and physiological or behavioral status, have been reported to potentially influence eDNA emission rates in salamanders [34]. Furthermore, Thalinger et al. [35] observed a positive correlation between eDNA detection and fish activity in seven fish species in freshwater.

There are inconsistencies among studies and discrepancies among target species with regard to the relationship between sampling time and eDNA concentration. Nocturnal carp eDNA concentrations increased 500-fold at night when fish biomass only doubled at a feeding site [36]. In addition, eDNA concentrations of nocturnal riverine giant salamander [21,37] and tailed frog [21] did not differ between the daytime and nighttime.

Attempts have been made to use eDNA analysis to detect Japanese eel in rivers [6,38] and in the ocean [39]. For Japanese eels, there is a positive correlation between eDNA concentration and wet weight, body length [40]. The eDNA concentration increases 10–200× during spawning [39]. The degradation of eDNA also affects its detection. Water temperature has a significant positive influence on eDNA degradation in Japanese eel [41]. Individual differences in eDNA concentration have been detected among eels of the same size [40]. Eel ecophysiology and behavior markedly affect the eDNA release rate.

The Japanese eel is nocturnal. During the daytime, it hides in holes or mud in rivers or ponds [42,43]. In captivity, eels often remain in pipes during the day and leave pipes at night in search of food [44]. Light, water temperature, tidal cycles, the moon phase, and other

external environmental factors influence eel ecophysiology and behavior [44]. To accurately evaluate Japanese eel biomass through eDNA analysis, it is necessary to clarify whether there is diurnal variation in the eDNA concentration of eels.

The aim of the present study was to investigate factors influencing Japanese eel eDNA concentrations. Clarifying such factors could facilitate the development of appropriate tools for the reliable estimation of the biomass of Japanese eel and other species using eDNA. To this end, we performed tank experiments focusing on two points. First, we assessed the effects of base pair sequences with probe region polymorphism on eDNA detection (Experiment 1). Second, we evaluated the influences of diel rhythm, activity, and individual differences in eDNA release rate on eDNA concentration (Experiment 2).

## Materials and methods

### Ethics statement

This study was conducted in accordance with the guidelines of the Regulation on Animal Experimentation of Kyoto University, Kyoto, Japan. No fish or other animals were harmed in any of the experiments performed herein. No ethical approval was required for the experimental procedure due to the common consumption of cultured eel. All 20 eels were housed for use in further research.

### Experiment 1: Effects of polymorphism on eDNA detection

**Fish selection, experimental tank design, and water collection.** The effects of probe region base pair sequence polymorphism on eDNA detection were examined. Twenty Japanese eels (*Anguilla japonica*, SL = 494 ± 18 mm, W = 159 ± 10 g) were purchased from a commercial provider (Unagikobo, Daigotsusyo Ltd., Shizuoka, Japan) and transferred to the Maizuru Fisheries Research Station on May 23, 2018. The animals cultured in Miyazaki (Japan) were received there as glass eels collected by local fishermen ~6 mo before purchase. No individual had any visible injury.

Six acrylic tanks (L × W × H = 900 mm × 300 mm × 300 mm; I.D. L × I.D. W = 890 mm × 290 mm; V = 100 L) were arranged in parallel, bleached with 0.1% (w/v) sodium hypochlorite, enclosed by a blue vinyl sheet on all sides, and covered on the bottom with a black vinyl sheet. They were filled to 21 cm depth with tap water (54 L) dechlorinated by a water purifier (Standard Neo; Marfied, Kanagawa, Japan). Aeration was provided by two air stones placed at each corner of the long axis of each tank.

Three cycles of a 1-week experiment were conducted from May 25 to June 14, 2018. Operators wore nitrile gloves during sampling and all procedures. Six individual eels that were not used in an experiment were placed one by one in each tank per cycle. The room temperature was 24 ± 1˚C. At 10:00 daily, 500 mL water was collected through a vinyl tube with a siphon from mid-depth of each tank and stored in a plastic bottle. The tanks were bleached with 0.1% (w/v) sodium hypochlorite and rinsed with tap water 1 d before each cycle. Before eel introduction, the DNA concentrations were determined for the tanks and found to be negative in the first cycle and positive in the second and third cycles. Nevertheless, the pre-eel DNA concentrations divided by those measured on the first day of each respective cycle were < 3.4% in the latter cycles. This foreign DNA contamination had a negligible impact on the subsequent quantitative analyses. Eighteen of the 20 eels were used in this experiment. The other two were kept in separate buckets for 1 h on June 14, 2018. Samples of the water surrounding all 20 eels were collected in 500-mL plastic bottles and used in the base pair sequence analyses. The plastic bottles were bleached with 0.1% (w/v) sodium hypochlorite and prewashed twice with sampling water before sample collection. After sampling, small volumes of purified tap water were

added to the tanks at feeding and bottom cleaning. Individual eels were labeled with elastomer tags in preparation for the subsequent experiment.

**Water filtration and eDNA extraction.** Water filtration and eDNA extraction were performed according to the instructions in the Environmental DNA Sampling and Experiment Manual (v. 2.1) [45] with slight modification. Each 500-mL water sample was passed through an aspirator fitted with a glass fiber filter (GF/F; 0.7 µm pore size; 47-mm diameter; Whatman, Maidstone, UK). Each 500-mL distilled water sample was filtered once per sampling day and used as a blank control. After every filtration process, the filtration devices were bleached with 0.1% (w/v) sodium hypochlorite for 5 min, rinsed with tap water, and rinsed again with distilled water. The filters were wrapped in aluminum foil, placed in plastic bags, and stored at −20˚C until DNA extraction. The entire process from sampling to preservation was performed within 1 h.

The eDNA extraction was conducted with a DNeasy blood & tissue kit (Qiagen, Hilden, Germany). Each filter was placed in a Salivette tube (Sarstedt, Nümbrecht, Germany) and centrifuged at $5,000 \times g$ for 3 min. Then 420 µL of a solution comprising 20 µL Proteinase K, 200 µL Buffer AL, and 200 µL $H_2O$ was added to the filter. The tube was incubated at 56˚C for 30 min and the lysed DNA was collected by centrifugation at $5,000 \times g$ for 3 min. Then 200 µL *tris*-ethylenediaminetetraacetic buffer (TE buffer) was added to the filter and the liquid was collected by centrifugation at $5,000 \times g$ for 3 min. Then 200 µL Buffer AL and 600 µL ethanol were added to the liquid and the mixture was transferred to a spin column and centrifuged at $6,000 \times g$ for 1 min. Subsequently, 100 µL Buffer AE was eluted into the liquid according to the manufacturer's instructions and the mixture was preserved at −20˚C. All buffers (except TE), Proteinase K, and the spin columns from the DNA extraction kit were used in the eDNA extraction in the present study.

**PCR analysis.** The Japanese eel eDNA concentrations were quantified by qPCR in a LightCycler 96 system (Roche Diagnostics, Mannheim, Germany) as in a previous study [41]. The DNA from each target species was amplified using species-specific primers and probe sets targeting the mitochondrial D-loop region. The forward primer (primer F) Aja–Dlp–F was 5′–TACATTTAATGGAAAACAAGCATAAGCC–3′, the reverse primer (primer R) Aja–Dlp–R was 5′–CGTTAACATTACTCTGTCAACTTACCTG–3′, and the probe Aja–Dlp–P was 5'–FAM–ACCCATAAACTGATAAATAG–MGB–3'. The amplified length was expected to be 138 bp. The species-specificities of the primer/probe sets were confirmed by Kasai et al. [41].

Each PCR reaction included 900 nM forward and reverse (F/R) primers and 125 nM TaqMan Probe, 7.5 µL TaqMan Environmental Master Mix 2.0 (Thermo Fisher Scientific, Waltham, MA, USA), 0.075 µL AmpErase uracil *N*-glycosylase (Thermo Fisher Scientific, Waltham, MA, USA), and 2 µL DNA sample. The total reaction volume was adjusted to 15 µL with PCR-grade water (Roche Diagnostics, Basel, Switzerland). Dilutions containing $3 \times 10^1$–$3 \times 10^4$ copies per PCR tube were prepared and used as quantification standards. The qPCR conditions were as follows: 2 min at 50˚C, 10 min at 95˚C, 55 cycles of 15 s at 95˚C, and 60 s at 60˚C. There were three replicates each of all samples and standard DNAs. Three replicate negative controls containing PCR-grade water instead of template DNA were included in all PCR plates. For all PCR runs, the calibration curves' $R^2$ were > 0.99, and the ranges of the slope, Y-intercept, and PCR efficiency were −3.58 to −3.36, 39.55 to 40.28, and 0.90 to 0.99, respectively. None of the PCR-negative or blank controls was PCR-amplified. The amplified fragments were directly sequenced by a commercial sequencing service (No. SQ18F210091; Fasmac, Atsugi, Kanagawa, Japan). To mitigate the risk of carryover contamination, the pre-PCR and post-PCR experiments were conducted in separate rooms.

The DNA from each target individual in the first and second cycles was quantified by an intercalator method using the same species-specific primers as in the above-mentioned

TaqMan method. Each PCR reaction system included 900 nM forward and reverse (F/R) primers, 7.5 μL PowerUp SYBR Green Master Mix (Thermo Fisher Scientific, Waltham, MA, USA), and 2 μL DNA sample. The total reaction volume, standard dilutions, and replications were similar to those in the TaqMan method above. The qPCR conditions were as follows: 2 min at 50˚C, 2 min at 95˚C, 55 cycles of 15 s at 95˚C and 60 s at 60˚C, and melting (15 s at 95˚C, 60 s at 60˚C, 0.20˚C/s from 60˚C to 95˚C, and 15 s at 95˚C). For all PCR runs, the calibration curves' $R^2$ were > 0.99, and the ranges of the slope, Y-intercept, and PCR efficiency were −3.68 to −3.50, 38.38 to 39.18, and 0.87 to 0.93, respectively. None of the PCR-negative or blank controls were PCR-amplified.

## Experiment 2: Diel eDNA release rate and eel activity patterns

**Fish selection.** Here, the effects of diel activity and individual differences on the Japanese eel eDNA release rates were examined. Four individual eels (Aja-1, Aja -2, Aja -3, Aja -4; SL 494 ± 21 mm; W 155 ± 7 g; S1 Table) were selected from the 20 used in Experiment 1 (S2 Table). Eels with similar base pair sequences between primers F and R were chosen. All had the same sequence except for Aja-2 whose 35th base from the 5′ end of primer F was "A" rather than "C" and Aja-4 whose 48th base from the 5′ end of primer F was "A" rather than "G" between the primer F-3′ and probe (S2 Table).

**Experimental tank design and activity evaluation.** Four acrylic tanks were arranged in parallel as described for Experiment 1 (Fig 1). The experimental space was enclosed by a black curtain to minimize external visual disturbances. Purified tap water was stored in a 100-L tank. The water level and volumes were 21 cm and 54 L, respectively. The water exchange rate was 225 mL/min over six cycles per day. The aeration rate was 200 mL/min. Dou et al. [46] reported that glass eels are more active at 20˚C and 24˚C than they are at 15˚C. In the present study, the eels moved more actively at ≥ 24˚C than they did at 20˚C. Hence, the water temperature was maintained at 25.7 ± 0.7˚C. Fluorescent light was turned on at 05:00 and the light intensity was 402–493 Lx. The light was switched off at 17:00 and the light intensity declined to 0.01–0.09 Lx. No feeding or bottom cleaning was performed during the experiment.

A video camera (SEC-TF-N060WISC; Broadwatch, Osaka, Japan) was set in each tank to record eel activity. Nocturnal behavior was recorded using the infrared function of the camera. Eel activity was scored at 1-s intervals as follows: eel motionless in the pipe, 0 points; eel moving and caudal fin out of the pipe, 1 point; at least half but less than the entire eel body out of the pipe, 2 points; entire eel body out of the pipe, 3 points. Eel activity was evaluated by calculating the sum of scores at 1 h or 3 h before each water sampling. Eel activity at 1 h and 3 h were compared to make correlations between eDNA concentration and activity.

**Water collection and filtration.** An eel was introduced into each tank at 08:00 on July 23, 2018. Water samples were collected at 09:00 between July 23 and July 25, 2018 (acclimatization). Eight samplings were conducted between July 26 and July 27, 2018 at 06:00, 09:00, 12:00, and 15:00 (daytime) and at 18:00, 21:00, 00:00, and 03:00 (nighttime). Water was collected in three 500-mL plastic bottles as for Experiment 1. Five hundred milliliters of water was collected from the tank in each plastic bottle before introducing an eel on July 20. These samples were used as blank controls.

Water filtration, eDNA extraction, and qPCR analysis by the TaqMan method using species-specific primers and probe sets were performed in the same manner as for Experiment 1. For all PCR runs, the calibration curves' $R^2$ were > 0.99. The ranges of the slope, Y-intercept, and PCR efficiency were −3.60 to −3.31, 39.12 to 40.34, and 0.90 to 1.00, respectively (S3 Table). None of the PCR-negative or blank controls was PCR-amplified.

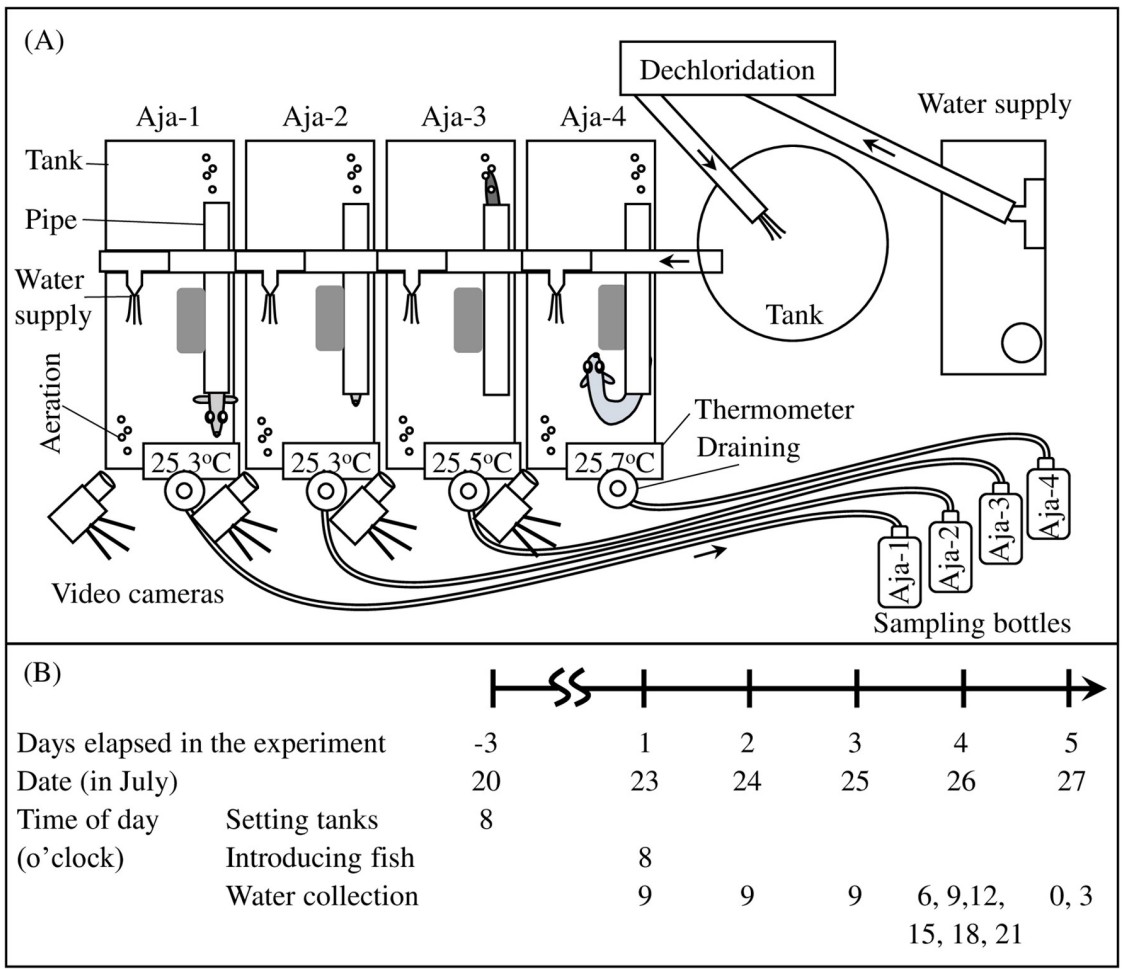

**Fig 1. (A) Schematic drawing of the experimental tank system and (B) time schedule.** Arrows in (A): Water flow direction.

**Data analysis.** All statistical analyses were performed in R v. 3.6.3 [47]. The diurnal and nocturnal eDNA concentration and activity were compared with a two-tailed, paired Student's *t*-test. Normality was tested by Shapiro's test, and then the homoscedasticity was verified by var-test for *t*-test and Bartlett's test for multiple comparisons (S4 Table). During acclimatization, the daily variation in eDNA concentration at 09:00 between July 23 and 26, 2018 were analyzed by one-way ANOVA with Tukey's HSD multiple comparisons test. During acclimatization, the individual variations in eDNA concentration at 09:00 between July 24 and 26, 2018 were analyzed by the Kruskal-Wallis and Dunn-Bonferroni multiple comparisons tests. Diurnal and nocturnal eDNA concentration and activity over time and among individuals were evaluated by one-way ANOVA and Tukey's HSD multiple comparisons test. Coefficients of variation among individuals, samplings, and PCR replicates were determined and compared by the Kruskal-Wallis and Dunn-Bonferroni multiple comparisons tests. Correlations between eDNA concentration and activity for each individual were estimated by linear regression (95% confidence and prediction limits) using the "lm" package in R v. 3.6.3.

## Results

### Experiment 1: Effects of polymorphism on eDNA detection

One out of 20 bases in the probe region was different in two of 20 individuals tested (S4 Table). Variant-1 (first cycle on May 25–31, 2018; Fig 2A) presented with a genetic mismatch in the middle of the probe region. Variant-2 (second cycle on June 1–7, 2018; Fig 2B) had a mismatch at the third base from the 3′ end of the probe region (Table 1). In terms of the eDNA variation within an individual experiment in 1 week, the variant eDNA concentrations were consistently low or undetected. In contrast, other individuals fluctuated between high and low eDNA concentrations (Fig 2A–2C). The fluorescence endpoints of the qPCR amplification curves decreased in the order of control, Variant-2, Variant-1 on the first day, Variant-1 on the second or later day, and negative control (Fig 2D; $Ct$ = 23.58 ± 0.07, 25.28 ± 0.04, 27.32 ± 0.23, and non-detectable (ND), respectively). Variant-1 had a positive eDNA concentration only immediately after it was introduced to the tank. Thereafter, its eDNA concentration was negative (Fig 2A).

By the intercalator method, the variant eDNA concentration was detected and was consistently as high as in the other individuals (Fig 2E and 2F). The fluorescence endpoints of the qPCR amplification curves were similar among the control, Variant-2, Variant-1 on the first day, and Variant-1 on the second or later day, and extremely low in the negative control (Fig 2G; $Ct$ = 20.81 ± 0.16, 21.58 ± 0.07, 22.25 ± 0.07, 20.35 ± 0.20, and ND, respectively). The melting temperature (Tm) was 74.08 ± 0.46˚C (S5 Table).

### Experiment 2: Diel eDNA release rate and eel activity patterns

During acclimatization, the eDNA concentration was significantly higher on the first day after fish introduction than it was by day four for all individuals except Aja-1 ($df$ = 3, Aja-1: $p$ = 0.99, Aja-2: $p$ = 0.02, Aja-3: $p$ < 0.01, Aja-4: $p$ < 0.01; Table 2). There was no significant difference between the second and fourth days for any individual ($df$ = 3, Aja-1: $p$ = 0.98, Aja-2: $p$ = 0.06, Aja-3: $p$ = 0.99, Aja-4: $p$ = 0.57; Table 2). Inter-individual variation in the eDNA concentration was significantly higher for Aja-4 than it was for Aja-1 and Aja-3 between the second and fourth days of acclimatization ($df$ = 3, $p$ < 0.01; Table 2). The eDNA concentration was significantly higher in Aja-4 than it was in the others ($df$ = 3, $F$ = 8.86, $p$ < 0.02; Fig 3). The activity level of Aja-3 was significantly higher than those of Aja-2 and Aja-4 ($df$ = 3, $F$ = 3.72, $p$ = 0.02; Fig 3).

Observation of the diel changes during the main experiment indicated that the eDNA concentration ($df$ = 3, $p$ = 0.03) and activity ($df$ = 3, $p$ < 0.01) during the nighttime (18:00, 21:00, 00:00, and 03:00) were significantly higher than those during the daytime (06:00, 09:00, 12:00, and 15:00; Fig 3). The activity at 21:00 was significantly higher than that at 12:00 ($df$ = 7, $F$ = 2.83, $p$ = 0.03; Fig 3). The eDNA concentration was minimal during the daytime, peaked around sunset, and decreased significantly by nighttime ($df$ = 7, Aja-1: $F$ = 16.2, $p$ < 0.05, Aja-2: $F$ = 122.6, $p$ < 0.04, Aja-3: $F$ = 74.8, $p$ < 0.02, Aja-4: $F$ = 35.4, $p$ < 0.03; Fig 3). Peak eDNA occurred between 15:00 and 21:00 and varied among individuals (Fig 3).

The timing at which the eels actively moved outside the pipes (activity score = 3) also differed among individuals between 15:00 and 21:00 (around sunset) and at 03:00 (predawn) and 6:00 (dawn) (Fig 3). Both the eDNA concentration and activity increased in Aja-4 at 09:00 (Fig 3D). The standard deviation for the eDNA concentration at 15:00 was very large for Aja-1 (Fig 3A).

There were inter-individual differences in the correlation between eDNA concentration and activity 1 h before water sampling. These correlations were very weakly positive for Aja-1

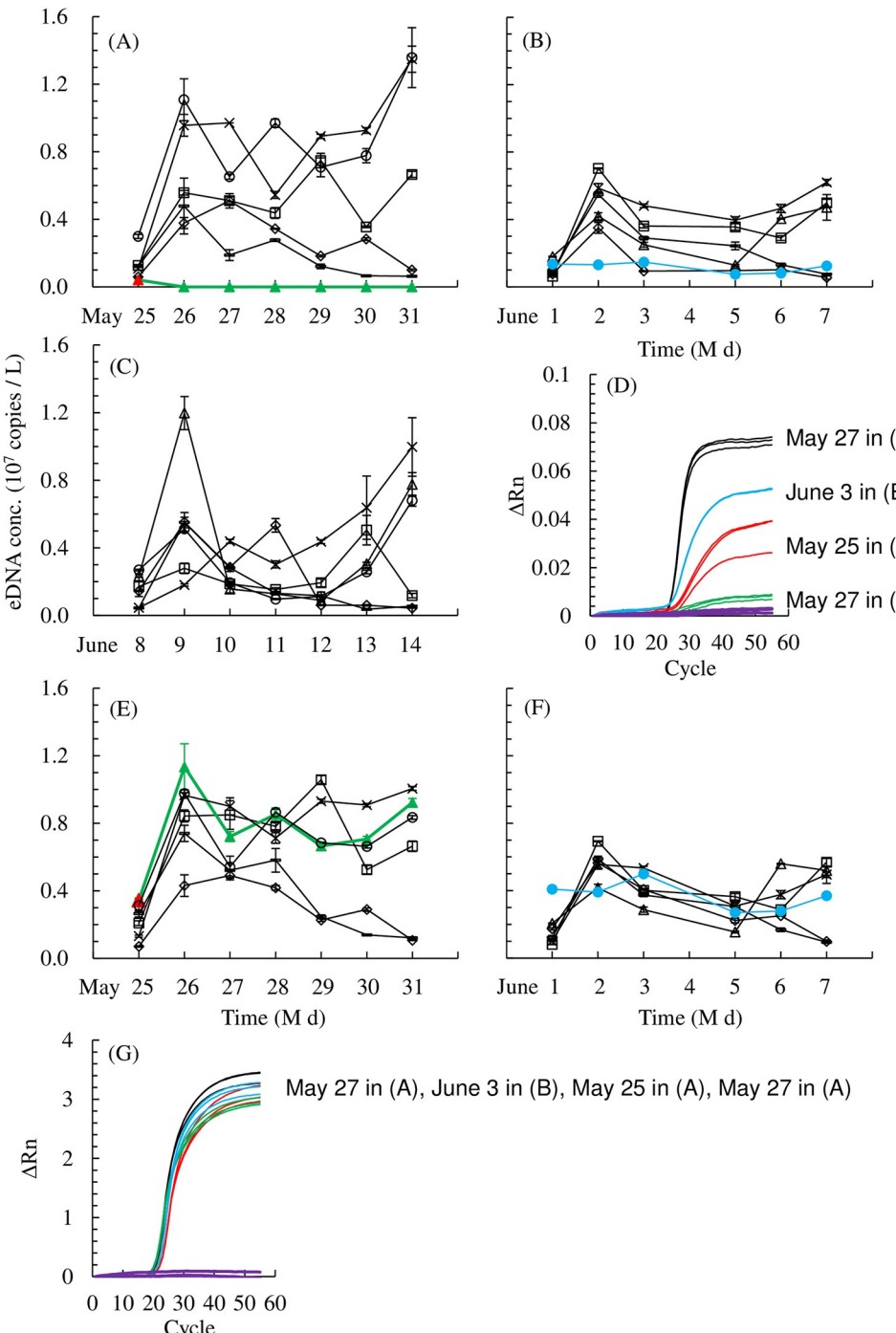

**Fig 2.** (A), (B), (C) Inter-individual variation in eDNA concentration (eDNA conc.) in 18 individual Japanese eels and (E), (F) in 12 individual Japanese eels, and (D), (G) amplification curves. (A), (B), (C), (D) the TaqMan method using species-specific primers and probe sets. (E), (F), (G) an intercalator method with SYBR Green. Black: Controls; red and green: Variant-1 (first day; second or later day of the first cycle); blue: Variant-2 (second cycle); purple: Negative controls; square: A-1; diamond: B-1; triangle: C-1; cross: A-2; bar: B-2; circle: C-2. Error bars represent standard deviations (SD) of PCR replication.

**Table 1. Probe region sequences.**

| Sample | Sequence of probe region[*] |
|---|---|
| Non variant | ACCCATAAACTGATAAATAG |
| Variant-1 | ACCCATAAA**T**TGATAAATAG |
| Variant-2 | ACCCATAAACTGATAAA**C**AG |

[*]Red: Base in polymorphism.

**Table 2. Variation in eDNA concentration (eDNA conc.) during acclimatization.**

| Sample No.[*2] | | eDNA conc. ($10^4$ copies/L)[*1] | | | |
|---|---|---|---|---|---|
| | Date | 23-Jul | 24-Jul | 25-Jul | 26-Jul |
| Aja-1[B] | | 25.0 ±3.4[b] | 17.3 ±0.9[b] | 98.0 ±30.1[a] | 22.1 ±2.8[b] |
| Aja-2[AB] | | 62.3 ±11.9[a] | 55.3 ±6.8[ab] | 50.9 ±10.3[ab] | 33.1 ±5.7[b] |
| Aja-3[B] | | 79.7 ±12.7[a] | 35.9 ±2.6[b] | 21.4 ±3.9[b] | 37.6 ±3.4[b] |
| Aja-4[A] | | 669.7 ±22.7[a] | 178.7 ±41.2[b] | 71.5 ±14.9[c] | 215.1 ±45.3[b] |

[*1]Average ± SD. Values within same row having different superscript letters are significantly different ($df = 3$, Aja-1: $F = 19.2$, $p < 0.01$, Aja-2: $F = 5.76$, $p = 0.02$, Aja-3: $F = 38.7$, $p < 0.01$, Aja-4: $F = 187.3$, $p < 0.01$, one-way ANOVA with Tukey's HSD multiple comparisons test).

[*2]Values within the same column having different superscript capital letters are significantly different ($df = 3$, $p < 0.01$; Kruskal-Wallis and Dunn-Bonferroni multiple comparisons tests).

and Aja-2 and positive for Aja-3 and Aja-4 ($R^2$ = 0.072, 0.023, 0.30, and 0.56, respectively; Fig 4). The correlation for the activity data 1 h before water sampling was considerably higher than that at 3 h before water sampling in Aja-2, Aja-3, and Aja-4 ($R^2$ = 0.12, 0.008, 0.13, and 0.43 in Aja-1, Aja-2, Aja-3, and Aja-4, respectively; S1 Fig).

The coefficient of variation among the individual eDNA concentrations was significantly larger than those for the sampling and PCR measurement replications ($df = 2$, $p < 0.01$; Fig 5). Therefore, the fluctuations in eDNA concentration shown in Table 2 and Fig 3 were derived mainly from individual variations.

## Discussion

### Genetic polymorphism may hinder eDNA studies

The environmental DNA of a target species is specifically detectable when there are a sufficient number of interspecific base pair mismatches [48]. Intraspecific mutations and polymorphisms affect eDNA detection by qPCR analysis using species-specific primers/probes [17,18]. It is reported that eDNA detection is more influenced by base pair mismatches in the primer region than in the probe region [18]. Mismatches in the probe region nonetheless affect the estimation of eDNA concentration [49]. The DNA amplicon is certainly amplified in the subsequent steps once the templates are annealed by the primers, because the sequences of DNA amplicons completely match the sequences in the priming sites of the primers. However, mismatches of sequences between probe sets and templates are not eliminated in the subsequent amplification steps. Therefore, the negative effects of polymorphs in probe regions on DNA amplification are potentially considerable.

Individual rearing of Japanese eel in Experiment 1 confirmed that base sequence polymorphism in the probe region substantially modulate the eDNA concentration. The endpoint fluorescence levels of the qPCR-amplified curves were lower than those of the others (Fig 2D). The

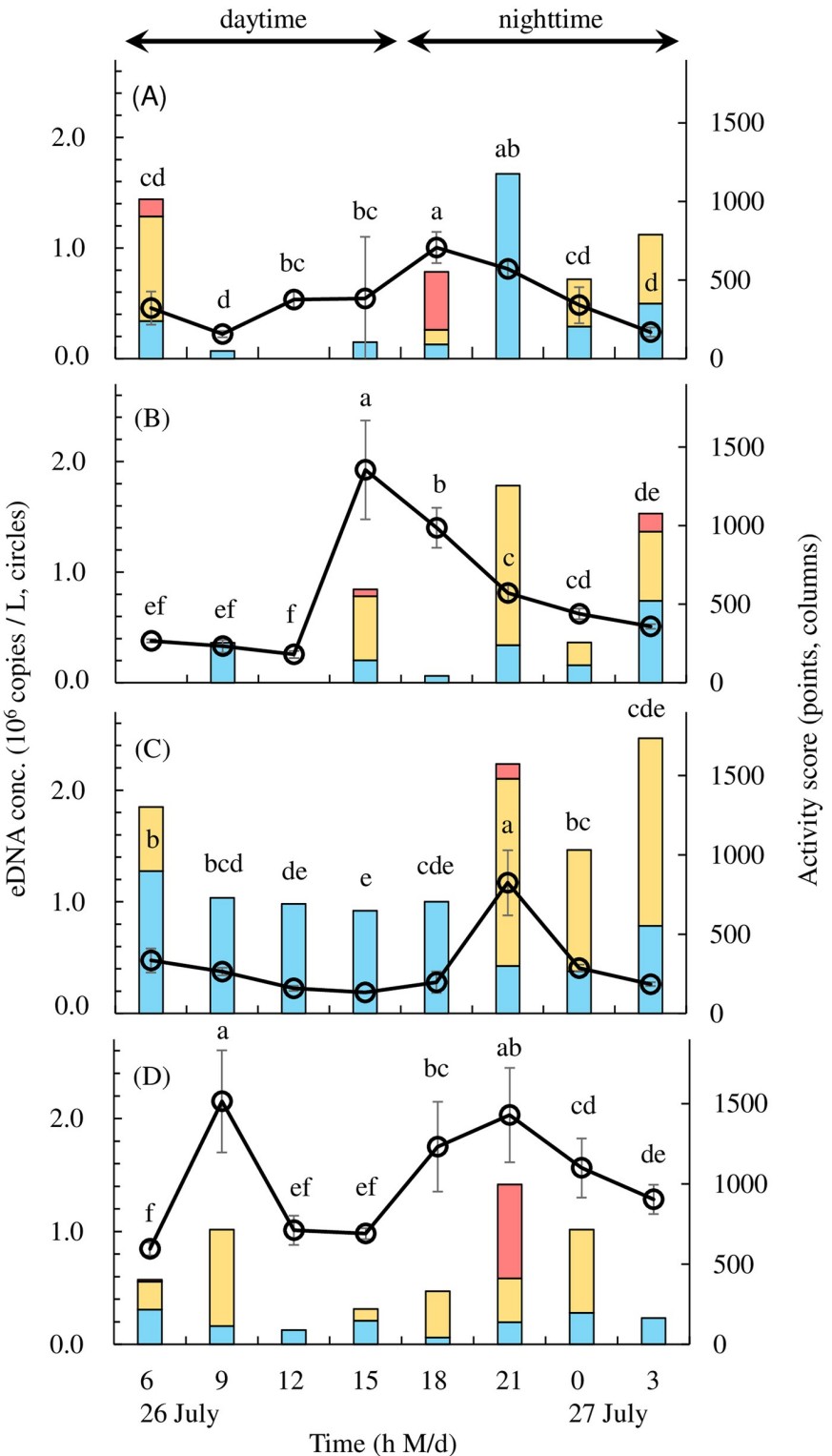

**Fig 3. Diel changes in eDNA concentration (eDNA conc.) (circles; left axis) and activity (columns, blue: 1, yellow: 2, red: 3; right axis) in Japanese eels.** (A) Aja-1, (B) Aja-2, (C) Aja-3, (D) Aja-4. There were significant differences in eDNA concentration ($df = 3$, $p = 0.03$) and activity ($df = 3$, $p < 0.01$) between daytime and nighttime (two-tailed, paired Student's $t$-test). There were significant differences in eDNA concentration among times of day ($df = 7$, $p < 0.05$, one-way ANOVA with Tukey's HSD multiple comparisons test). Error bars indicate standard deviations.

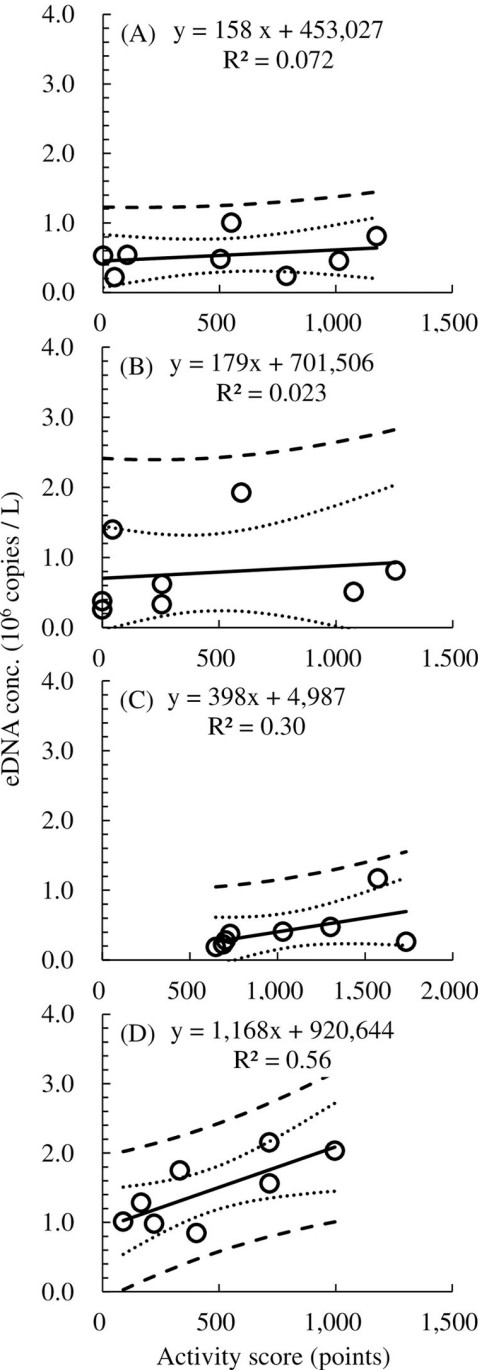

**Fig 4. Correlation between eDNA concentration (eDNA conc.) and activity score 1 h before water sampling.** (A) Aja-1, (B) Aja-2, (C) Aja-3, (D) Aja-4. Dotted lines indicate 95% confidence limits. Dashed lines indicate 95% prediction limits.

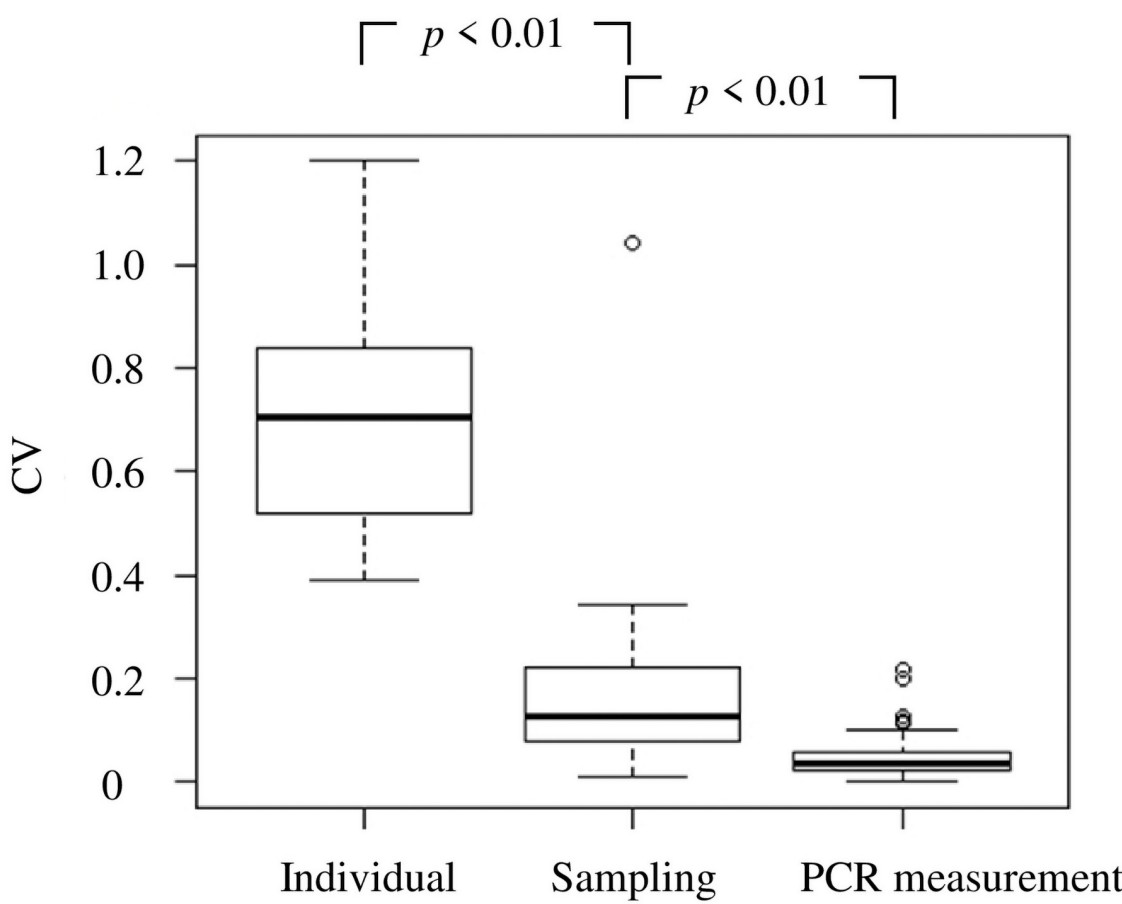

**Fig 5. Coefficients of variation (CV) in individual, sampling, and PCR measurement replications.** The CV significantly differed ($df = 2$, $p < 0.01$, Kruskal-Wallis and Dunn-Bonferroni multiple comparisons tests).

eDNA concentrations of the variants were consistently estimated to be lower than those of the others (Fig 2A and 2B). Earlier studies reported flattened PCR amplification curves [17,48]. The eDNA concentration of the eel with a genetic mismatch in the middle of the probe region (Variant-1) was lower than that of the eel with a mismatch in the side of the probe region (Variant-2) (Fig 2A and 2B). In the former case, the eDNA concentration was low but positive immediately after the eel was introduced to the tank (Fig 2A). From the second day onward, however, its eDNA concentration was negative (Fig 2A). We have no reasonable explanation for this phenomenon, which is a subject to be addressed in future research.

In general, a probe is hybridized to complementary DNA during qPCR annealing [49,50], and then is degraded and separated by the 5′-to-3′ exonuclease activity of DNA polymerase during target-specific DNA amplification. This reaction is responsible for fluorescence emission [49,51]. The melting temperature (Tm, ˚C) decreases and the probe hybridization weakens in response to base pair mismatches [49]. In particular, the use of MGB accentuates the difference in Tm due to single nucleotide polymorphisms (SNPs), and the probe is less likely to anneal to the target region. We propose that the fluorescence decreases and the eDNA concentration is underestimated when probe hybridization is interrupted by a base pair mismatch in the probe region. Probe hybridization would decrease when the mismatch occurs in the middle of the probe region. A similar polymorphism was detected in the probe region, and

weak fluorescence was observed in mitochondrial *cytochrome b* of jack mackerel (Takahashi et al., *unpublished data*).

Using the intercalator method with SYBR Green in our study, eDNA concentrations and the endpoint fluorescence levels of the variants were consistently as high, and their *Ct* values were as low as those in the non-variant individuals (Fig 2E–2G). These results confirmed that the total amount of DNA extracted from the filters in Variant-1 and -2 was equivalent to that of the other individuals, and inhibitors were absent from these samples.

Some researchers have used Japanese eel primers/probe sets targeting mitochondrial 16S ribosomal RNA (rRNA) (153 bp [6], 107 bp [39], and 154 bp [40]). Here, we used primers/probe sets targeting 138 bp of the mitochondrial D-loop region. These primers/probe sets have also been used in Kasai et al. [7], where a nationwide eel distribution survey in Japan was conducted. Kasai et al. [41] carefully checked the specificity in the set of primers used in the present study, as follows. The *A. japonica* sequence was compared with those of 14 congeneric subspecies in order to confirm specific amplification. The primer/probe sequences were consistent with the target species consensus sequence based on 857 individuals of *A. japonica* samples. In this manner, base pair mismatches caused by intraspecific genetic variation were minimized. However, a base sequence mismatch in the probe region was detected in two of the 20 individual eels. Individual tank experiments in our study using these primers/probe sets revealed that polymorphism in the probe region influenced eDNA detection considerably. It is necessary to design primers/probe sets selecting regions with relatively low intraspecific polymorphism, which would facilitate the precise measurement of eDNA concentrations in each individual.

D-loop regions have higher rates of polymorphism than other mitochondrial regions. Nevertheless, other mitochondrial regions can also be polymorphed. The risk of overestimation following PCR amplification of non-native con-generic species' DNA is increased by the selection of regions that generally have lower mutation rates than the D-loop region, whereas the risk of underestimation caused by the primer/probe mismatch could be reduced. Therefore, it is a trade-off between overestimation and underestimation where to select a target region when applying eDNA technologies to ecological research.

The target species biomass could be underestimated in marine and riverine DNA surveys when the variant is mixed at the survey point. We have no data concerning the ratio of variants in the natural eel population. From the results of our study, 10% of the total population were variants, which is unlikely to bias the presence–absence data when each habitat hosts a large enough number of individuals. When eDNA concentrations are applied to estimate biomass, such a polymorphism is likely to lead to underestimation. If some habitats host only a few individuals, then such a polymorphism might lead to false negatives. Therefore, we recommend that the sequence of the target species inhabiting the survey area should be verified in advance depending on the purpose of the eDNA study.

## Inter-individual difference and daily rhythm of activity affect eDNA emission

The concentration of eDNA often fluctuates soon after the introduction of fish in a tank. For instance, it took 3 days in the case of bluegill sunfish (*Lepomis macrochirus*) [29] and 6 days in the case of common carp (*Cyprinus carpio*) [20] until eDNA concentrations stabilized in the still water of the tanks. In the present experiment, although eDNA concentrations on the fourth day were significantly lower than those on the first day in three individuals, the difference was non-significant between the second and fourth days in all individuals. Therefore, eDNA concentrations seem to have stabilized from the second day onward, perhaps due to the

running water system used in this study. This was advantageous for comparing inter-individual, as well as diel, differences in eDNA concentrations.

The coefficients of variation in the individual eDNA concentration measurements were significantly larger than those for the sampling and PCR replications (Fig 5). The fluctuations in eDNA concentration might be the result of individual variations in the eDNA release rate. Positive correlations have been reported between fish size and eDNA concentration [29,30,40]. As all individuals of this study were equivalent in size, the variation would be caused by factors other than biomass differences.

Here, we focused on the diurnal activity patterns as putative factors affecting the eDNA concentrations. Japanese eels are nocturnal [42,43]. Glass eels often remain in pipes during the daytime and emerge immediately after the light source is switched off [44]. We used elvers whose diurnal rhythm was assumed to be the same as that of glass eels. Their nighttime activity was significantly greater than their daytime activity (Fig 3). All individuals were more active at 21:00 than they were at 12:00. Thus, the Japanese eels were active at sunset and non-active at noon.

Carp eDNA concentration increased at night in accordance with their nocturnal activity [36]. In contrast, the eDNA concentrations of salamanders and frogs did not markedly differ between daytime and nighttime [21,37]. Ghosal et al. [36] suggested that a nocturnal increase in eDNA concentration could be explained by increases in carp biomass and activity as the fish gather in feeding areas at night. We focused on the nighttime increase in Japanese eel activity and detected changes in eDNA concentration by examining individual eels of the same size. The nocturnal eDNA concentrations were higher than the diurnal ones. Therefore, the observed variations in eDNA concentration coincided with the nocturnal activity of this species.

A positive correlation between eDNA concentration and fish activity has been reported in seven freshwater fishes [35]. In the present study, eDNA concentration and activity were also positively correlated. The activity 1 h before water sampling was more strongly correlated with the eDNA concentration than the activity 3 h before water sampling (Figs 4 and S1 Fig). Moreover, the correlation between eDNA concentration and activity varied among individuals. We propose that the factors affecting eDNA concentration include activity and individual differences in basal metabolism. In the metabolic process, the eels seem to release substantial amounts of eDNA.

Japanese eels move actively at dawn and dusk [44]. In the present study, individual differences in active time were observed. Aja-1 moved actively at dawn (06:00) and dusk (18:00) whereas Aja-2 moved less actively at these times. Relative differences in individual activity were also detected between midnight (00:00) and predawn (03:00) (Fig 3). The activity and eDNA concentration at 09:00 were higher for Aja-4 than they were for the other eels (Fig 3). Throughout the day, the eDNA concentration was the lowest in the daytime, highest at dusk, and gradually decreased thereafter. Peak eDNA concentrations varied among individuals between 15:00 and 21:00. During this time interval, the eels actively moved outside the pipes. The high eel activity at that time coincided with the maximum eDNA concentration, suggesting that a large amount of mucus and other substances containing their DNA was shed from their body.

The activity level of Aja-3 was significantly higher than those of Aja-2 and Aja-4. In contrast, the eDNA concentration of Aja-4 was significantly higher than those of Aja-1 and Aja-3 during acclimatization between the second and fourth days and those of all other eels during the main experiment. Hence, eDNA release is not always higher in more active individuals. Possible eDNA sources include urine, mucus, saliva, carcasses, and feces [11,12]. Furthermore, the aquatic eDNA state may be free, cellular, or particle-bound [13]. In lakes, eDNA

concentrations may be higher near the bottom than the surface [23]. The same is true for sea-water [52]. Consequently, eDNA distribution may be irregular. DNA patches in the water may also affect the eDNA concentration. Patchy aquatic eDNA distribution might account for the large standard deviation of the eDNA concentration in Aja-1 at 15:00 (Fig 3A).

## Conclusions and perspectives

We demonstrated that a base pair mismatch in the probe region may lead to the underestimation of eDNA detection in Japanese eel. The problem of genetic polymorphism, as we have shown, is likely to occur in the eDNA of other species. The present study also showed that the Japanese eel diel rhythm and activity affect their eDNA emissions. There was a positive correlation between eDNA concentration and eel activity and both were relatively higher in the nighttime. Therefore, water collection at sunset maximizes riverine eel detection in eDNA-based distribution surveys. Moreover, eDNA release rates differ among individual eels. These findings may well contribute to improve the accuracy and precision of estimating eel abundance and distribution using eDNA. This should be an important step to protect and rebuild the depleted stock of this species. However, the sample size used in the present study was limited; hence, extending the survey scale is preferable before confidently upscaling eDNA technology in the field. The effect of mismatch in priming sites of primers is also worth evaluating to improve the efficiency of eDNA detection in field surveys. The internal and external environments of target species also influence the variation in eDNA concentration. To improve the accuracy of biomass estimation, further investigation is required to identify the other factors affecting eDNA concentration and the interactions that may occur among those factors.

## Supporting information

**S1 Fig. Correlation between eDNA concentration and activity score 3 h before water sampling.**
(TIF)

**S1 Table. Length and weight of samples.**
(XLSX)

**S2 Table. Sequence results for 20 *Anguilla japonica* samples.**
(XLSX)

**S3 Table. Parameters of standard curves for qPCR measurements on each sampling date.**
(XLSX)

**S4 Table. *P*-values of Shapiro's test, var-test, and Bartlett's *t*-test.**
(XLSX)

**S5 Table. Melting temperatures (Tm) of standard and each samples.**
(XLSX)

**S6 Table.** (a) Data of eDNA concentration (eDNA conc.), (b) amplification curves, and (c) parameters of the standard curve for each qPCR measurement shown in Figs 1 and 2. (a), (d) Data of eDNA concentration (eDNA conc.), (b), (e) amplification curves, and (c), (f) parameters of the standard curve for each qPCR measurement shown in Fig 2.
(XLSX)

**S7 Table. eDNA concentrations in Table 2, Figs 3–5 and S1 Fig.**
(XLSX)

**S8 Table. Activity score used in Figs 3 and 4 and S1 Fig.**
(XLSX)

## Acknowledgments

The authors thank Mr. Tatsuki Toya of Kyoto University for his assistance with the measurement of eel body length and wet weight. The authors also thank the students and staff of Maizuru Fisheries Research Station of Kyoto University for their assistance with our experiments. We would also like to thank Dr. Hideyuki Doi (University of Hyogo) and two anonymous reviewers for their constructive comments that helped us to substantially improve the quality of the manuscript.

## Author Contributions

**Conceptualization:** Sayaka Takahashi.

**Data curation:** Sayaka Takahashi, Reiji Masuda.

**Formal analysis:** Sayaka Takahashi.

**Funding acquisition:** Reiji Masuda, Akihide Kasai.

**Investigation:** Sayaka Takahashi, Shingo Takada, Reiji Masuda.

**Methodology:** Sayaka Takahashi, Shingo Takada, Hiroki Yamanaka, Reiji Masuda, Akihide Kasai.

**Project administration:** Sayaka Takahashi, Reiji Masuda.

**Resources:** Reiji Masuda.

**Validation:** Sayaka Takahashi, Hiroki Yamanaka, Akihide Kasai.

**Visualization:** Sayaka Takahashi.

**Writing – original draft:** Sayaka Takahashi.

**Writing – review & editing:** Sayaka Takahashi, Shingo Takada, Hiroki Yamanaka, Reiji Masuda, Akihide Kasai.

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
