## [Decision Letter · Decision Letter 0]

26 Jan 2021

PONE-D-20-38269

Intraspecific genetic variability and diurnal activity affect environmental DNA detection in Japanese eel

PLOS ONE

Dear Dr. Takahashi,

Thank you for submitting your manuscript to PLOS ONE. After careful consideration, we feel that it has merit but does not fully meet PLOS ONE’s publication criteria as it currently stands. Therefore, we invite you to submit a revised version of the manuscript that addresses the points raised during the review process.

We look forward to receiving your revised manuscript.

Kind regards,

Hideyuki Doi

Academic Editor

PLOS ONE

Additional Editor Comments:

I got the recommendations and comments from two expert reviewers on the field. The both reviewer agree that the manuscript is interesting but the both reviewers raised serious concern to the markers in this study and some other concerns. I totally share their comments. Therefore, I regrettably inform you that I should reject the manuscript in current form, but with substantial revisions according to the reviewers comments, especially the marker issue with adding further experiment results, I can invite you to submit a revised version of the manuscript.

Journal Requirements:

2.) In your Methods section, please include a comment about the state of the animals following this research. Were they euthanized or housed for use in further research? If any animals were sacrificed by the authors, please include the method of euthanasia and describe any efforts that were undertaken to reduce animal suffering.

3.) Thank you for including your ethics statement:  "This study was conducted in accordance with the guidelines of the Regulation on Animal Experimentation of Kyoto University, Kyoto, Japan. No fish or other animals were harmed in any of the experiments performed herein.".   

Please amend your current ethics statement to confirm that your named ethics committee specifically approved this study.

For additional information about PLOS ONE submissions requirements for ethics oversight of animal work, please refer to http://journals.plos.org/plosone/s/submission-guidelines#loc-animal-research  

Reviewers' comments:

Reviewer's Responses to Questions

**Comments to the Author**

1. Is the manuscript technically sound, and do the data support the conclusions?

Reviewer #1: Partly

Reviewer #2: Yes

2. Has the statistical analysis been performed appropriately and rigorously? 

Reviewer #1: I Don't Know

Reviewer #2: Yes

3. Have the authors made all data underlying the findings in their manuscript fully available?

Reviewer #1: Yes

Reviewer #2: Yes

4. Is the manuscript presented in an intelligible fashion and written in standard English?

Reviewer #1: No

Reviewer #2: Yes

5. Review Comments to the Author

Reviewer #1: I have reviewed the manuscript “Intraspecific genetic variability and diurnal activity affect environmental DNA detection in Japanese eel” by Sayaka Takahashi et al. This study found the mismatch on the priming region of the species-specific probe for Japanese eel caused to under-estimate of eDNA concentration. Besides, it was suggested that the diel activity rhythm affect eDNA concentration. These findings offer an important message for future eDNA studies. However, I have several serious concerns about the experimental design and manuscript in its present form. I like the data the authors present, but it falls short of a stand-alone paper about the effect of probe mismatch for detection and the diurnal activity on eDNA concentration. Much more experimental work and careful discussion are required for this to exist as a stand-alone paper.

<major comments="">

L67-69: I am very confused because I understood that the purpose of this study was to examine whether the mismatches of the specie-specific probe leads to a decrease in detection (L98-99).

L59-60, 70-73: There are duplications of text content between paragraphs.

L88-93: To clarify the significance of this study, it is necessary to explain what problems may occur if there is diurnal variation in eDNA concentration of Japanese eel. We agree that there are contradictions in results among previous studies, but you should not discuss based on studies using some taxa with very different ecological characteristics.

L75, 100: What is "individual differences"? (ex. size, figure, sex, physiological state, nutrition condition etc.) In this study, only the difference in activity was considered.

L155-162: The amounts of extraction reagents written in the manuscript are different from those written in the Environmental DNA Sampling and Experiment Manual (v. 2.1). Is the first centrifugation a newly added step? Also, how was the filtrate from the first centrifugation processed (discarded or kept) ?

L170-176: The D-loop region has a higher mutation rate, so intraspecific genetic diversity is more likely to accumulate. Even though other species-specific primers which were designed in other regions with lower mutation rate than D-loop have been published, why did you not use them?

L174: The Japanese eel-specific probe used in this study had MGB fluorescent dye. The use of MGB accentuates the difference in Tm due to SNPs, so the probe is less likely to anneal to target region when there is a mismatch in the priming site. Please add discussions about the possibility that the MGB emphasized the effect of probe mismatches on DNA detection in this study.

L226-230: I don't think the method of scoring the activity used in this study is appropriate. How was the score for each activity determined? Why didn't you use the amount of movement distance per unit of time or the number of respirations or fin movements? I think the current scoring system underestimates the amount of action to get out from a pipe and forage for food.

L359-361: I am very confused because there is a lack of explanation about Figure 2D. Why only five amplification curves are shown in figure 2D? Also, why are the collection dates of the samples different? (Not standardized to a specific day, ex. the first day, of each experiment cycle)

L365-368: From the second day, the eDNA concentrations of all eels without probe mismatch increased significantly. Considering this result, it is natural to assume that the eels with mismatch also likely released more eDNA on the second day than the first day. Please add a discussion about the reason why no DNA was detected after the second day, even though there may have been more eDNA.

L383-386: To design a more robust species-specific primers/probe, all obtained sequences should be used and compared to select SNP-free regions. Consensus sequences are not simply calculated based on the number and frequency of mutations. In addition, it is important that the selection of the target region. There is more mutation in the mitochondrial D-loop region because it has a higher mutation rate than other regions. The selection of cytb, 16S and 12S etc. decrease the risk of unexpected mismatch between primer/probe sequence and the target region.

L386-388: Japanese eels spawn near the Mariana Islands, and juveniles are randomly carried by ocean currents to the coast of Japan. Therefore, it is not surprising that 20 individuals purchased from the same aquaculture provider have different haplotypes.

L428: I couldn't understand what this sentence was referring to. Please add a more detail explanation.

L438-439: Why is it limited to mucus? I am very confused.

L453-454: I think that the Japanese eel "ecophysiology" was not examined. In addition, it is difficult to say that it revealed the effect of "behavior" on DNA concentration because this study only evaluated whether the eels were out of the pipe and/or moved their caudal fins.

Fig.4: Please use same Y-axis range among graphs.

Table S1: Which primers were used for Sanger sequencing of each individual? If the DNA amplicon was sequenced using the same primers as qPCR, you will not be able to find mismatches in the priming sites of the primers. Additionally, what does the ( ) on the reverse primer site indicate?

・This study does not explore the effect of probe mismatch for detection enough. While probe mismatches are likely to inhibit the detection of the target sequence (I agree with this), if the authors are serious about making a contribution to this issue, they should have examined the impact of sequence mismatches on detection rate using a known concentration DNAs.

・In experiment 1, authors cannot exclude the possibility that the eel with the sequence mutations released less eDNA than other eels. To investigate this possibility, it is highly recommended to quantify DNA concentrations by both of TaqMan and intercalater methods (ex. SYBR Green) to clearly show that sequence mismatch of TaqMan probe is inhibiting detection. Although the authors emphasize the high species-specificity of the primers/probe (in methods and discussion), it is not so important in this study because only eels were kept in tank.</major>

Reviewer #2: Manuscript PONE-D-20-38269

This manuscript asses the effect of intraspecific polymorphism as well as diel activity rhythm on eDNA detection/quantification in Japanese eels (Anguilla japonica). In a set of experimental tanks, the authors found that the presence of DNA polymorphisms located within the probe region of some individuals (2 eels), tends to underestimate the quantification of eDNA when qPCR method is employed. Moreover, a correlation between eDNA concentration and eel activity was found, being higher at nighttime.

The manuscript is well written and the methodology is mostly clear, although further editing and additional information is necessary. I found this paper interesting and I think that has the potential to increase our knowledge about eDNA detection and quantification. However, I found several issues that should be addressed before the acceptance of the manuscript.

- In my opinion, the choice of marker was poorly justified. It is well known that Dloop is one of the most polymorphic region within the mitochondrial DNA. Do the authors expect to have such an impact in the case of other, less variable, markers. On top of that, it will be important to report the level of polymorphism (at the probe region) on natural populations of eels. Thus, this will allow the reader to identify the magnitude of the bias when screening natural populations.

- Related with the previous point, I was surprised that the authors didn´t test additionally the use of degenerated probes in order to surpass the underestimation of eDNA (or the lack of detection).

- I was wondering if the total amount of DNA extracted from the filters was independently quantified using other methods (e.g. Qubit). This would allow to discard that underestimated samples (e.g. “Mutants”) didn´t have less amount of total DNA than the others.

- I was also wondering if inhibitor´s test were performed, particularly for those samples were no detection was possible. It is expected that Negative samples spiked-in with positive controls should amplified in the absence of inhibitors.

- I think is more correct to use “polymorphism” instead of “mutation” all over the text. The “mutants” that are reported here, seem to be common polymorphisms in natural populations.

- Despite the fact the authors performed 3 replicates per sample for eDNA quantification/detection, I was wondering why they use only 1 biological replicate. Each time, only one water sample was taken. The authors should discuss this point and it will be important to know if the level of variation within tank at each time-point was significant.

Minor comments:

L28: intraspecific

L41: polymorphic sites at the probe region

L97: to identify factors

L117: Miyazaki (Japan)

L125: from May 25, to June 14, 2018

L126: unused?

L205: In my opinion, Table 1 could be shown as supplementary table.

L235: 23, and July

6. PLOS authors have the option to publish the peer review history of their article (what does this mean?). If published, this will include your full peer review and any attached files.

Reviewer #1: No

Reviewer #2: No

---

## [Author Response · Author response to Decision Letter 0]

13 Apr 2021

April 13, 2021

PhD. Hideyuki Doi,

Academic Editor,

PLOS ONE

Dear Editor:

I, along with my co-authors, would like to re-submit the attached manuscript entitled newly “Intraspecific genetic variability and diurnal activity affect environmental DNA detection in Japanese eel” as a research article. (‘Response to Reviewers’, ‘Revised Manuscript with Track Changes’, ‘Manuscript’ figures 5; table 2; Supporting Information 9). The paper was co-authored by Shingo Takada, Hiroki Yamanaka, Reiji Masuda, Akihide Kasai.

The manuscript has been carefully rechecked and appropriate changes have been made according to the reviewer’s suggestion. The responses to their comments have been prepared and are attached herewith.

We thank you and the reviewers for your thoughtful suggestions and insights, which have enriched the manuscript and produced a more balanced and better account of the research. We hope that the revised manuscript is now suitable for publication in your journal.

We are not allowed to deposit our laboratory protocols in protocols.io, and the protocols were fully described in our manuscript.

I look forward to your reply.

Sincerely,

Sayaka Takahashi

Oita Junior College of Horticulture

Chiyomachi, Oita, 870-8658, Japan

Phone: +81-97-535-0201, Fax: +81-97-540-6509, e-mail: takahashi@oitatandai.ac.jp

Date: Jan 26 2021 07:23PM

To: "Sayaka Takahashi" tsayaka@life.shimane-u.ac.jp

From: "PLOS ONE" plosone@plos.org

Subject: PLOS ONE Decision: Revision required [PONE-D-20-38269]

PONE-D-20-38269

Intraspecific genetic variability and diurnal activity affect environmental DNA detection in Japanese eel

PLOS ONE

Dear Dr. Takahashi,

Thank you for submitting your manuscript to PLOS ONE. After careful consideration, we feel that it has merit but does not fully meet PLOS ONE’s publication criteria as it currently stands. Therefore, we invite you to submit a revised version of the manuscript that addresses the points raised during the review process.

・ A rebuttal letter that responds to each point raised by the academic editor and reviewer(s). You should upload this letter as a separate file labeled 'Response to Reviewers'.

・ A marked-up copy of your manuscript that highlights changes made to the original version. You should upload this as a separate file labeled 'Revised Manuscript with Track Changes'.

・ An unmarked version of your revised paper without tracked changes. You should upload this as a separate file labeled 'Manuscript'.

We look forward to receiving your revised manuscript.

Kind regards,

Hideyuki Doi

Academic Editor

PLOS ONE

Additional Editor Comments:

I got the recommendations and comments from two expert reviewers on the field. The both reviewer agree that the manuscript is interesting but the both reviewers raised serious concern to the markers in this study and some other concerns. I totally share their comments. Therefore, I regrettably inform you that I should reject the manuscript in current form, but with substantial revisions according to the reviewers comments, especially the marker issue with adding further experiment results, I can invite you to submit a revised version of the manuscript.

Journal Requirements:

Our reply: we revised file names, according to the PLOS ONE style templates.

2.) In your Methods section, please include a comment about the state of the animals following this research. Were they euthanized or housed for use in further research? If any animals were sacrificed by the authors, please include the method of euthanasia and describe any efforts that were undertaken to reduce animal suffering.

Our reply: We housed eels for use in further research. We added “All 20 eels were housed for use in further research” in the Materials section (L 118-119).

3.) Thank you for including your ethics statement: "This study was conducted in accordance with the guidelines of the Regulation on Animal Experimentation of Kyoto University, Kyoto, Japan. No fish or other animals were harmed in any of the experiments performed herein.". 

Please amend your current ethics statement to confirm that your named ethics committee specifically approved this study.

Our reply: In this study, no fish or other animals were harmed in any of the experiments performed herein, and we did not need ethical approval as long as we conduct a tank experiment in accordance with the guidelines of the Regulation on Animal Experimentation of Kyoto University. We have added “No ethical approval was required for the experimental procedure due to the common consumption of cultured eel” in the Materials section (L 117-118).

For additional information about PLOS ONE submissions requirements for ethics oversight of animal work, please refer to http://journals.plos.org/plosone/s/submission-guidelines#loc-animal-research

Reviewers' comments:

Reviewer's Responses to Questions

Comments to the Author

1. Is the manuscript technically sound, and do the data support the conclusions?

Reviewer #1: Partly

Reviewer #2: Yes

2. Has the statistical analysis been performed appropriately and rigorously?

Reviewer #1: I Don't Know

Reviewer #2: Yes

3. Have the authors made all data underlying the findings in their manuscript fully available?

Reviewer #1: Yes

Reviewer #2: Yes

4. Is the manuscript presented in an intelligible fashion and written in standard English?

Reviewer #1: No

Reviewer #2: Yes

5. Review Comments to the Author

Reviewer #1: I have reviewed the manuscript “Intraspecific genetic variability and diurnal activity affect environmental DNA detection in Japanese eel” by Sayaka Takahashi et al. This study found the mismatch on the priming region of the species-specific probe for Japanese eel caused to under-estimate of eDNA concentration. Besides, it was suggested that the diel activity rhythm affect eDNA concentration. These findings offer an important message for future eDNA studies. However, I have several serious concerns about the experimental design and manuscript in its present form. I like the data the authors present, but it falls short of a stand-alone paper about the effect of probe mismatch for detection and the diurnal activity on eDNA concentration. Much more experimental work and careful discussion are required for this to exist as a stand-alone paper.

(Response)

Thank you for your interest in the contents and for your constructive comments. All comments provided have been very valuable in improving our manuscript. We extensively revised our manuscript, especially we added more experiment and careful discussion. In addition, we revised “mutation” as “polymorphism” in the text. We have used polymorphism as a term to represent the phenomenon of having a base pair mismatch in a primers/probe region and (genetic) variant as to represent an individual having such a base pair mismatch.

#R1-1: L67-69: I am very confused because I understood that the purpose of this study was to examine whether the mismatches of the specie-specific probe leads to a decrease in detection (L98-99).

Our reply: In quantitative PCR (qPCR) analysis, base pair mismatches between target species and probes are reported to reduce detection of DNA in human [26] or primers in fish [27]. Wilcox et al. [27] showed that eDNA detection of fish was decreased by base pair mismatches in primer regions rather than that in probe regions. These studies are different from our study in target species (human or fish) and findings (discussed in Discussion section: L 407-411). To avoid any confusion, we revised these sentences as “In quantitative PCR (qPCR) analysis using species-specific primers/probes, mutations and polymorphisms will affect the eDNA detection rate. Base pair mismatches between target species and probes have been reported to reduce the detection of DNA in humans [26] or primers in fish [27]” (L 69-74).

#R1-2: L59-60, 70-73: There are duplications of text content between paragraphs.

Our reply: We thank the reviewer for identifying this duplicative impression. Actually, the former represents those in the field and the latter in tank experiments. We made sentences to be more explicit in the revised manuscript (L 61-63 and L 75-78, respectively).

#R1-3: L88-93: To clarify the significance of this study, it is necessary to explain what problems may occur if there is diurnal variation in eDNA concentration of Japanese eel. We agree that there are contradictions in results among previous studies, but you should not discuss based on studies using some taxa with very different ecological characteristics.

Our reply: As per the reviewer’s suggestion, we added “To accurately evaluate Japanese eel biomass through eDNA analysis, it is necessary to clarify whether there is diurnal variation in the eDNA concentration of eels” at the end of this paragraph (L 100-102). We are focusing on “nocturnal/diurnal habits” beyond taxa with different ecological characteristics.

#R1-4: L75, 100: What is "individual differences"? (ex. size, figure, sex, physiological state, nutrition condition etc.) In this study, only the difference in activity was considered.

Our reply: We thank the reviewer for pointing it out. We added “and the suspected factors are listed as metabolic rate, stress, life stage, and physiological or behavioral state [34]” at the end of this sentence (L 81-83). We revised the reference of [13] as a new reference [34] (it was a careless mistake). We described “individual differences” as “individual differences in eDNA release rate on eDNA concentration” in the latter sentence (L 110).

#R1-5: L155-162: The amounts of extraction reagents written in the manuscript are different from those written in the Environmental DNA Sampling and Experiment Manual (v. 2.1). Is the first centrifugation a newly added step? Also, how was the filtrate from the first centrifugation processed (discarded or kept) ?

Our reply: As the reviewer pointing it out, the amounts of extraction reagents written in the manuscript are sometimes different from those written in the Manual. Therefore, we added “with a slight modification” at the end of this sentence (L 159-160). All modifications that we made have been written in the revised manuscript. The first centrifugation is a newly added step. The first centrifugation (removing extra water from filters, which had absorbed water during the preservation at -20 °C) is needed to lyse DNA effectively in the following process. The filtrate from the first centrifugation was kept in the lower part of a Salivette tube.

#R1-6: L170-176: The D-loop region has a higher mutation rate, so intraspecific genetic diversity is more likely to accumulate. Even though other species-specific primers which were designed in other regions with lower mutation rate than D-loop have been published, why did you not use them?

Our reply: As the reviewer pointing it out, the D-loop region is known to have a higher polymorphic rate than other mitochondrial regions. However, the other mitochondrial regions can also be polymorphed. Actually, we detected similar polymorphisms in the probe region targeting mitochondrial cytochrome b of jack mackerel (L 436-439). We took polymorphism as a subject to be generally careful about, irrespective of mitochondrial regions. In this study, we used the same primers/probe sets as Kasai et al [36] who surveyed nationwide eel distribution in Japan. The specificity has been carefully checked based on 857 individuals of A. japonica tissue samples, as well as all other congeneric species which could potentially be detected due to food consumption or release into rivers (L 456-458). Other species-specific primers, which were designed in other regions with lower polymorphic rate than D-loop, have been published (e.g., 16S: Watanabe et al (2004), or Cytb: Takahara et al (2020) L&O methods). However, these primer sets have not been validated in the original studies for their specificity against all of the non-target closely related species in the same genus listed in our main manuscript using genomic DNA samples. Therefore, we used the one used in Kasai et al [36].

#R1-7: L174: The Japanese eel-specific probe used in this study had MGB fluorescent dye. The use of MGB accentuates the difference in Tm due to SNPs, so the probe is less likely to anneal to target region when there is a mismatch in the priming site. Please add discussions about the possibility that the MGB emphasized the effect of probe mismatches on DNA detection in this study.

Our reply: As per the reviewer suggestion, we added discussions about the possibility that the MGB emphasized the effect of probe mismatches on DNA detection (L 430-432); “In particular, the use of MGB accentuates the difference in Tm due to single nucleotide polymorphisms (SNPs), and the probe is less likely to anneal to the target region when there is a mismatch in the priming site”. We have conducted an additional quantification of eDNA samples by intercalator method (SYBR Green) (L 207-221), and added the results (L 326-331) and the discussion (L 430-432, 440-445).

#R1-8: L226-230: I don't think the method of scoring the activity used in this study is appropriate. How was the score for each activity determined? Why didn't you use the amount of movement distance per unit of time or the number of respirations or fin movements? I think the current scoring system underestimates the amount of action to get out from a pipe and forage for food.

Our reply: The amount of movement distance per unit of time, the number of respirations or fin movements were not used because 1) we needed to set a pipe to reduce the stress of each eel at the tank experiment, and respirations and fin movements were not possible to be counted when it was in a pipe; 2) eels moved only occasionally, and thus movement distance per hour would be very low; and 3) we found that the activity amount of an eel in a pipe was lower than that of the eel moving caudal fin out of the pipe or that of the body being out of the pipe. Therefore, we determined the score for each activity in ascending order for convenience sake (L 255-259).

#R1-9: L359-361: I am very confused because there is a lack of explanation about Figure 2D. Why only five amplification curves are shown in figure 2D? Also, why are the collection dates of the samples different? (Not standardized to a specific day, ex. the first day, of each experiment cycle)

Our reply: We obtained too many amplification curves to be shown in a figure, so we selected representative values of them (Fig 2-1D). We standardized the collection dates of the samples to the first day after acclimatization (May 27 and June 3) in order to avoid the influence of contaminations (L 143-148), except for May 25, when eDNA concentration of Polymorphism-1 was positive (Fig 2-1A, 2-1B).

#R1-10: L365-368: From the second day, the eDNA concentrations of all eels without probe mismatch increased significantly. Considering this result, it is natural to assume that the eels with mismatch also likely released more eDNA on the second day than the first day. Please add a discussion about the reason why no DNA was detected after the second day, even though there may have been more eDNA.

Our reply: We agree with the reviewer. The result by an intercalator method with SYBR Green also showed that Variant-1 released more eDNA on the second day than the first day (Fig 2-2A). We do not know the cause of this and added a sentence (L 423-424); “We have no reasonable explanation for this phenomenon, which is a subject to be solved in future research”.

#R1-11: L383-386: To design a more robust species-specific primers/probe, all obtained sequences should be used and compared to select SNP-free regions. Consensus sequences are not simply calculated based on the number and frequency of mutations. In addition, it is important that the selection of the target region. There is more mutation in the mitochondrial D-loop region because it has a higher mutation rate than other regions. The selection of cytb, 16S and 12S etc. decrease the risk of unexpected mismatch between primer/probe sequence and the target region.

Our reply: As explained in the reply above, the D-loop region is known to have a higher polymorphic rate than other mitochondrial regions. However, other mitochondrial regions still have intraspecific variation. From the observation that we detected similar polymorphisms in the probe region using mitochondrial cytochrome b of jack mackerel (L 436-439, [17]), we focused on the risk of an unexpected mismatch from primer/probe sequence template as a subject to be generally careful about, irrespective of mitochondrial regions. In this study, we carefully checked the intraspecific variation of the primers/probe sets, as described above (#R1-6). We showed this result for the purpose of contributing to design a more robust species-specific assay.

#R1-12: L386-388: Japanese eels spawn near the Mariana Islands, and juveniles are randomly carried by ocean currents to the coast of Japan. Therefore, it is not surprising that 20 individuals purchased from the same aquaculture provider have different haplotypes.

Our reply: We thought that haplotypes of the eels obtained from a single provider who purchased juveniles and cultivated in one time would be more similar than those obtained from several providers who cultivated at various timing. Nevertheless, the above is speculative and so we delated “simultaneously obtained from the same aquaculture provider” (L 460-461).

#R1-13: L428: I couldn't understand what this sentence was referring to. Please add a more detail explanation.

Our reply: To avoid any confusion, we revised “In the latter process” as “In the metabolic process” (L 522).

#R1-14: L438-439: Why is it limited to mucus? I am very confused.

Our reply: To avoid any confusion, we have added “and other substances” after “mucus” (L 533).

#R1-15: L453-454: I think that the Japanese eel "ecophysiology" was not examined. In addition, it is difficult to say that it revealed the effect of "behavior" on DNA concentration because this study only evaluated whether the eels were out of the pipe and/or moved their caudal fins.

Our reply: To avoid any confusion, we revised “ecophysiology and behavior” as “diel rhythm and activity” (L 550-551).

#R1-16: Fig.4: Please use same Y-axis range among graphs.

Our reply: As per the reviewer's suggestion, we revised Y-axis range in Fig 4 (and S1 Fig).

#R1-17: Table S1: Which primers were used for Sanger sequencing of each individual? If the DNA amplicon was sequenced using the same primers as qPCR, you will not be able to find mismatches in the priming sites of the primers. Additionally, what does the ( ) on the reverse primer site indicate?

Our reply: We thank the reviewer for this constructive comment. We used the same primers as qPCR for Sanger sequencing. Around the end of primer R-3', base peaks overlapped and another base was occasionally judged, which was shown as “()”. Such phenomenon is generally found around the end of primer R-3'. On the other hand, presenting primer base sequences would not make much sense as the reviewer pointed out. Therefore, they were deleted from the previous S1 Table to be the new S2 Table.

#R1-18: ・This study does not explore the effect of probe mismatch for detection enough. While probe mismatches are likely to inhibit the detection of the target sequence (I agree with this), if the authors are serious about making a contribution to this issue, they should have examined the impact of sequence mismatches on detection rate using a known concentration DNAs.

Our reply: We agree with this comment by the reviewer. However, the suggested experiment would not be possible, because we have no tissue samples of the eels left. We have eDNA samples from eel tank water which we have analized the sequence, though the eDNA concentration isn’t high enough to prepare a series of DNA samples with different concentrations. We think it is important to show the effect of sequence mismatches in the probe annealing region in this first report, though we have indirectly confirmed the effect of probe mismatching by conducting eDNA quantification by SYBR Green method that doesn’t emply the probe as above. Even though we didn’t directry assess the effect of the mismatches using degenerated probes we recognize the importance of this study which elucidates this phenomenon in an individual tank experiment for the first time.

#R1-19: ・In experiment 1, authors cannot exclude the possibility that the eel with the sequence mutations released less eDNA than other eels. To investigate this possibility, it is highly recommended to quantify DNA concentrations by both of TaqMan and intercalater methods (ex. SYBR Green) to clearly show that sequence mismatch of TaqMan probe is inhibiting detection. Although the authors emphasize the high species-specificity of the primers/probe (in methods and discussion), it is not so important in this study because only eels were kept in tank.

Our reply: This DNA region is a non-coding region, so it is unlikely that the eel with the sequence mismatch released less eDNA than other eels. The difference among individuals was naturally large and we observed this in the other species than eels (ex. jack mackerel and striped knifejaw: Oplegnathus fasciatus), although we do not know if these species had base mismatches in the D-loop region or not. We performed an additional experiment to quantify DNA concentrations by an intercalator method with SYBR Green (L 207-221). As a result, concentrations of the polymorphic DNAs were demonstrated to be as high as the others, implying that the sequence mismatch of the TaqMan probe is inhibiting eDNA detection (L 326-331; Fig 2-2). 

Reviewer #2: Manuscript PONE-D-20-38269

This manuscript asses the effect of intraspecific polymorphism as well as diel activity rhythm on eDNA detection/quantification in Japanese eels (Anguilla japonica). In a set of experimental tanks, the authors found that the presence of DNA polymorphisms located within the probe region of some individuals (2 eels), tends to underestimate the quantification of eDNA when qPCR method is employed. Moreover, a correlation between eDNA concentration and eel activity was found, being higher at nighttime.

The manuscript is well written and the methodology is mostly clear, although further editing and additional information is necessary. I found this paper interesting and I think that has the potential to increase our knowledge about eDNA detection and quantification. However, I found several issues that should be addressed before the acceptance of the manuscript.

(Response)

Thank you for your interest in the contents and for your constructive comments. All comments provided have been very valuable in improving our manuscript. We extensively revised our manuscript. Specifically, we performed an additional experiment and the Discussion has been enriched.

#R2-1: - In my opinion, the choice of marker was poorly justified. It is well known that Dloop is one of the most polymorphic region within the mitochondrial DNA. Do the authors expect to have such an impact in the case of other, less variable, markers. On top of that, it will be important to report the level of polymorphism (at the probe region) on natural populations of eels. Thus, this will allow the reader to identify the magnitude of the bias when screening natural populations.

Our reply: As the reviewer pointed out, the D-loop region is known to have a higher polymorphic rate than other mitochondrial regions. However, other regions can also be polymorphed. Indeed, we detected similar polymorphisms in the probe region targeting mitochondrial cytochrome b in jack mackerel (L 436-439). We took up the polymorphism as a subject to be careful about, irrespective of the mitochondrial regions. In this study, we used the same primers/probe sets as Kasai et al [36], who conducted a nationwide eel distribution survey in Japan using these primers. They checked the specificity of the primers based on 857 individuals of A. japonica tissue samples (> 93% base pair sequence matching with the probe region sequence [36]) and all the congeneric species that would be potentially detected in rivers due to food consumption or release of introduced individuals (L 456-458). We have no data concerning the level of polymorphism in natural populations. As we detected two variants out of 20 purchased individuals and juveniles should have been well randomized from the spawning site, the level might be around 10 %. This would be negligible when eDNA is applied to presence/absent evaluation when the population is abundant, whereas inducing slight underestimation in the quantification. When the population is small, it can also be a cause of false negative data even in presence/absence surveys. These inferences have been included in the revised Discussion of the manuscript (L 463-471).

#R2-2: - Related with the previous point, I was surprised that the authors didn´t test additionally the use of degenerated probes in order to surpass the underestimation of eDNA (or the lack of detection).

Our reply: Instead of testing the use of degenerated probes, we did an additional experiment to quantify DNA concentrations by an intercalator method with SYBR Green (L 207-221). The concentration of the polymorphic DNAs was confirmed to be as high as the others (L 326-327; Fig 2-1, 2-2). This result clearly showed that a probe base mismatch was the cause of the underestimation of eDNA (or the lack of detection).

#R2-3: - I was wondering if the total amount of DNA extracted from the filters was independently quantified using other methods (e.g. Qubit). This would allow to discard that underestimated samples (e.g. “Mutants”) didn´t have less amount of total DNA than the others.

Our reply: As mentioned above (#R2-2), the concentration of the polymorphic DNAs was equivalent to the others according to the intercalator method with SYBR Green (L 326-327; Fig 2-1, 2-2). We added “These results confirmed that the total amount of DNA extracted from the filters in Variant-1 and -2 was equivalent to that of the other individuals” in the Discussion section (L 442-445).

#R2-4: - I was also wondering if inhibitor´s test were performed, particularly for those samples were no detection was possible. It is expected that Negative samples spiked-in with positive controls should amplified in the absence of inhibitors.

Our reply: As mentioned above (#R2-2), the concentration of the polymorphic DNAs was equivalent to the others in the intercalator method with SYBR Green (L 326-327; Fig 2-1, 2-2). This result excludes the possibility of containing inhibitors. We added “ These results confirmed that the total amount of DNA extracted from the filters in Variant-1 and -2 was equivalent to that of the other individuals, and inhibitors were absent from these samples” in the Discussion section (L 442-445).

#R2-5: - I think is more correct to use “polymorphism” instead of “mutation” all over the text. The “mutants” that are reported here, seem to be common polymorphisms in natural populations.

Our reply: We thank the reviewer for giving us an opportunity to reconsider this term. As the reviewer pointed out, mutation may not be the ideal term in this manuscript. In the revised manuscript, we have used polymorphism as a term to represent the phenomenon of having a base pair mismatch in a primers/probe region and (genetic) variant to represent an individual having such a base pair mismatch. 

#R2-6: - Despite the fact the authors performed 3 replicates per sample for eDNA quantification/detection, I was wondering why they use only 1 biological replicate. Each time, only one water sample was taken. The authors should discuss this point and it will be important to know if the level of variation within tank at each time-point was significant.

Our reply: We collected three 500-mL plastic bottles of water per sampling in Experiment 2, and discussed; “The coefficients of variation in the individual eDNA concentration measurements were significantly larger than those for the sampling and PCR replications (Fig 5). The fluctuations in eDNA concentration might be the result of individual variations in the eDNA release rate” (L 494-497). Here, the most important result is; “eDNA concentrations were constantly low in polymorphic samples by TaqMan method, though eDNA concentrations increased in polymorphic samples by an intercalator method with SYBR Green.”

Minor comments:

L28: intraspecific (� L 29)

L41: polymorphic sites at the probe region (� L 42-43)

L97: to identify factors (� L 106)

L117: Miyazaki (Japan) (� L 129)

L125: from May 25, to June 14, 2018 (� L 137-138)

L126: unused? (� L 139)

L205: In my opinion, Table 1 could be shown as supplementary table. (� L 234-235)

L235: 23, and July (� L 264-265)

Our reply: Thank you for pointing them out. As per your suggestion, we revised them, and “unused eels” as “individuals of eels that had not experienced an experiment” (L 139), and Table 1 as S1 Table (L 234-235).

6. PLOS authors have the option to publish the peer review history of their article (what does this mean?). If published, this will include your full peer review and any attached files.

Do you want your identity to be public for this peer review? For information about this choice, including consent withdrawal, please see our Privacy Policy.

Reviewer #1: No

Reviewer #2: No

---

## [Decision Letter · Decision Letter 1]

17 May 2021

PONE-D-20-38269R1

Intraspecific genetic variability and diurnal activity affect environmental DNA detection in Japanese eel

PLOS ONE

Dear Dr. Takahashi,

Thank you for submitting your manuscript to PLOS ONE. After careful consideration, we feel that it has merit but does not fully meet PLOS ONE’s publication criteria as it currently stands. Therefore, we invite you to submit a revised version of the manuscript that addresses the points raised during the review process.

I got the recommendations and comments from a previous reviewers. The reviewer agree that the manuscript is improved but suggested that more experimental work and careful discussion are necessary for examining the effect of mismatch between probe and target sequence on detection efficiency and provided other major/minor comments. I totally share these comments. Therefore, I can invite you to submit a revised version of the manuscript that addresses the points raised by the reviewer.

We look forward to receiving your revised manuscript.

Kind regards,

Hideyuki Doi

Academic Editor

PLOS ONE

Additional Editor Comments (if provided):

I got the recommendations and comments from a previous reviewers. The reviewer agree that the manuscript is improved but suggested that more experimental work and careful discussion are necessary for examining the effect of mismatch between probe and target sequence on detection efficiency and provided other major/minor comments. I totally share these comments. Therefore, I can invite you to submit a revised version of the manuscript that addresses the points raised by the reviewer.

Reviewers' comments:

Reviewer's Responses to Questions

**Comments to the Author**

1. If the authors have adequately addressed your comments raised in a previous round of review and you feel that this manuscript is now acceptable for publication, you may indicate that here to bypass the “Comments to the Author” section, enter your conflict of interest statement in the “Confidential to Editor” section, and submit your "Accept" recommendation.

Reviewer #1: (No Response)

2. Is the manuscript technically sound, and do the data support the conclusions?

Reviewer #1: Partly

3. Has the statistical analysis been performed appropriately and rigorously? 

Reviewer #1: I Don't Know

4. Have the authors made all data underlying the findings in their manuscript fully available?

Reviewer #1: Yes

5. Is the manuscript presented in an intelligible fashion and written in standard English?

Reviewer #1: Yes

6. Review Comments to the Author

Reviewer #1: I have re-reviewed the manuscript “Intraspecific genetic variability and diurnal activity affect environmental DNA detection in Japanese eel” by Sayaka Takahashi et al. The revised manuscript addresses some of the major concerns; however, several more experimental work and careful discussion are necessary for examining the effect of mismatch between probe and target sequence on detection efficiency. I offer a few major and minor suggestions for improvement below.

・The current title and text may cause the reader to misunderstand. This study suggested that mismatches between probes and target sequences, not genetic diversity, affect the efficiency of eDNA detection. In fact, SNP at a site other than the priming sites of primers and probes do not affect the estimation of eDNA concentration by qPCR.

・In the responses to the reviewer, the authors used the same primers with qPCR for the Sanger sequencing to check the presence of SNPs. Again, “if the DNA amplicon was sequenced using the same primers as qPCR, you will not be able to find mismatches in the priming sites of the primers” because the priming site of the primers is replaced by the primer sequence. I think, this study did not examine whether there was a sequence mismatch in the primer positions (not probe). According to Kasai et al (2020), the priming sites of the primers sometimes have SNP. Thus, in my opinion, without confirming that there is no mismatch in the priming sites of the primers, the results in this study cannot properly understand and discussed.

<introduction>

・The overall impression is redundant because the point of the argument in each paragraph is not clear, and similar content is written several times.

L55-56: Itakura et al. (2019) demonstrated that eDNA analysis allows us to reveal the spatial distribution, abundance, and biomass of Japanese eels at the river-basin scale.

L73-75: Is this true? I think, the relationship between eDNA concentration and biomass and/or the number of individuals is a major topic that has been examined by a great number of studies, and some of which have reported a strong relationship.

L79: “The eDNA concentration increases 10–200× during spawning [35].” → "[25]" ?

L90: Ghosal et al. (2018) reported that the carp eDNA concentration increased 500× at night when the fish biomass only doubled. However, this experiment was carried out under special experimental conditions. Thus, descriptions with generality should be avoided.

<materials and="" methods="">

L157, 169: “-20 °C” → “−20 °C”

L182: “Each PCR reaction system included” → “Each PCR reaction included”

L190: “There were three replicates per sample” → There were three replicates per sample “and standard DNAs”?

L198: “each target species” → each target individual ?

L174-176, L423-426: I still have a question as to why we did not use other species-specific primers and probe sets designed in other regions. Again, the specificity is not so important in this study because tank water was used in all experiments.

In the responses to the reviewers, the authors accepted that D-loop is more likely to accumulate intraspecific genetic diversity than other regions. The other mitochondrial regions can also be polymorphed, but the "mutation rate" vary widely among regions. I agree that the presence of polymorphisms is an issue to be generally careful. However, in the eDNA study, the risk of underestimation caused by the primer/probe mismatch can be reduced by selecting regions that generally have lower mutation rates than D-loop.

In my opinion, in this manuscript, the authors should include the discussion about the selection of the target region to reduce the risk of mismatches and the reason why the primer-probe set designed in the D-loop was used in this study.

<discussion>

L394-396: The eDNA concentration was different among samples. Thus, the comparison of Ct values here is meaningless.

・In my opinion, in all tables and Figures, eDNA concentration should be shown per volume of filtered water (500 mL) or per PCR template (2 µL).

・Please check the following points: (1) Are coloured letters allowed in PLOS One? (2) Are the legends inserted in the correct place in the manuscript?

Table1: Please add the Tm of the probes that are expected in each situation (Non variant, Vatiant-1, and Variant-2).

Fig2-1(D): What does the Y-axis indicate?

Fig2-1 and 2: To compare the eDNA concentration for each sample quantified by TaqMan (Fig. 2-1) and SYBR (Fig. 2-2), I recommend using sample-specific marks.

Fig.2-1(A), 2(A): The red and orange dots are difficult to distinguish, so I recommend changing the colour.

S2 table: “between 5'-primer F and probe” → “between 3'-primer F and probe” ?

“between probe and primer R-3'” → “between probe and primer R-5'” ?

“*Green: base different from sample 0525_A-1.” → What does 0525 mean?</discussion></materials></introduction>

7. PLOS authors have the option to publish the peer review history of their article (what does this mean?). If published, this will include your full peer review and any attached files.

Reviewer #1: No

---

## [Author Response · Author response to Decision Letter 1]

9 Jul 2021

July 9, 2021

PhD. Hideyuki Doi,

Academic Editor,

PLOS ONE

Dear Editor:

I, along with my co-authors, would like to re-submit the attached manuscript entitled newly “Intraspecific genetic variability and diurnal activity affect environmental DNA detection in Japanese eel” as a research article. (‘Response to Reviewers’, ‘Revised Manuscript with Track Changes’, ‘Manuscript’ figures 5; table 2; Supporting Information 9). The paper was co-authored by Shingo Takada, Hiroki Yamanaka, Reiji Masuda, Akihide Kasai.

The manuscript has been carefully rechecked and appropriate changes have been made according to the reviewer’s suggestion. The responses to the comments have been prepared and are attached herewith.

We thank you and the reviewer for your thoughtful suggestions and insights, which have enriched the manuscript. We hope that the revised manuscript is now suitable for publication in your journal.

We are not allowed to deposit our laboratory protocols in protocols.io, and the protocols were fully described in our manuscript.

I look forward to your reply.

Sincerely,

Sayaka Takahashi

Oita Junior College of Horticulture

Chiyomachi, Oita, 870-8658, Japan

Phone: +81-97-535-0201, Fax: +81-97-540-6509, e-mail: takahashi.sayaka.k35@kyoto-u.jp

PONE-D-20-38269R1

Intraspecific genetic variability and diurnal activity affect environmental DNA detection in Japanese eel

PLOS ONE

Dear Dr. Takahashi,

Thank you for submitting your manuscript to PLOS ONE. After careful consideration, we feel that it has merit but does not fully meet PLOS ONE’s publication criteria as it currently stands. Therefore, we invite you to submit a revised version of the manuscript that addresses the points raised during the review process.

I got the recommendations and comments from a previous reviewers. The reviewer agree that the manuscript is improved but suggested that more experimental work and careful discussion are necessary for examining the effect of mismatch between probe and target sequence on detection efficiency and provided other major/minor comments. I totally share these comments. Therefore, I can invite you to submit a revised version of the manuscript that addresses the points raised by the reviewer.

We look forward to receiving your revised manuscript.

Kind regards,

Hideyuki Doi

Academic Editor

PLOS ONE

Additional Editor Comments (if provided):

I got the recommendations and comments from a previous reviewers. The reviewer agree that the manuscript is improved but suggested that more experimental work and careful discussion are necessary for examining the effect of mismatch between probe and target sequence on detection efficiency and provided other major/minor comments. I totally share these comments. Therefore, I can invite you to submit a revised version of the manuscript that addresses the points raised by the reviewer.

Reviewers' comments:

Reviewer's Responses to Questions

Comments to the Author

1. If the authors have adequately addressed your comments raised in a previous round of review and you feel that this manuscript is now acceptable for publication, you may indicate that here to bypass the “Comments to the Author” section, enter your conflict of interest statement in the “Confidential to Editor” section, and submit your "Accept" recommendation.

Reviewer #1: (No Response)

2. Is the manuscript technically sound, and do the data support the conclusions?

Reviewer #1: Partly

3. Has the statistical analysis been performed appropriately and rigorously?

Reviewer #1: I Don't Know

4. Have the authors made all data underlying the findings in their manuscript fully available?

Reviewer #1: Yes

5. Is the manuscript presented in an intelligible fashion and written in standard English?

Reviewer #1: Yes

6. Review Comments to the Author

Reviewer #1: I have re-reviewed the manuscript “Intraspecific genetic variability and diurnal activity affect environmental DNA detection in Japanese eel” by Sayaka Takahashi et al. The revised manuscript addresses some of the major concerns; however, several more experimental work and careful discussion are necessary for examining the effect of mismatch between probe and target sequence on detection efficiency. I offer a few major and minor suggestions for improvement below.

Our reply: We thank the reviewer for the constructive comments, which have substantially improve the readability of the manuscript. 

・The current title and text may cause the reader to misunderstand. This study suggested that mismatches between probes and target sequences, not genetic diversity, affect the efficiency of eDNA detection. In fact, SNP at a site other than the priming sites of primers and probes do not affect the estimation of eDNA concentration by qPCR.

(Response)

We thank the reviewer for identifying the potential misunderstanding. The present result, i.e. less detection of eDNA in variants, indeed reflects the mismatches between probes and taget species as the reviewer suggested. However, essential finding of the study is that genetic variability would affect eDNA detection in the field even in a carefully designed set of primers and probe. We consider that there is a trade-off between speficity to target species and the risk of including variability in the probe region. This has been emphasized in the Discussion (Line 506–519) and we hope that our intention would be conveyed better in the revised manuscript. 

・In the responses to the reviewer, the authors used the same primers with qPCR for the Sanger sequencing to check the presence of SNPs. Again, “if the DNA amplicon was sequenced using the same primers as qPCR, you will not be able to find mismatches in the priming sites of the primers” because the priming site of the primers is replaced by the primer sequence. I think, this study did not examine whether there was a sequence mismatch in the primer positions (not probe). According to Kasai et al (2020), the priming sites of the primers sometimes have SNP. Thus, in my opinion, without confirming that there is no mismatch in the priming sites of the primers, the results in this study cannot properly understand and discussed.

(Response)

Although the presence of SNPs affects DNA amplification, the DNA amplicon is certainly amplified in the following steps once the templates are annealed by the primers, because the sequence of DNA amplicon completely matches the sequence in the priming sites of the primers. On the other hand, mismatches of sequences between probe and templates are not eliminated in the following amplification steps. Therefore, we consider that the negative effect of polymorphs in probe region on DNA amplification should be substantial.

As per the reviewer pointed out, we could not discuss whether there was a sequence mismatch of the primer region in our study without samples. We cannot show the information that completely wipes out the concern of the reviewer. However, eDNA amplification in Variants by SYBR Green method was mach more larger than that by TaqMan method. DNAs of Vatiant-1 and Variant-2 were amplified to the same amount as Non variants’ by using the same primers with qPCR in additional experiment by SYBR Green method. Therefore, the mismatches in the priming sites of the primers are not likely to have affected eDNA detection.

Nevertheless, we agree that it is also important to consider such a possibility, and thus it has been mentioned as a key consideration in the revised Conclusions and perspectives section (Line 608-611).

・The overall impression is redundant because the point of the argument in each paragraph is not clear, and similar content is written several times.

(Response)

We thank the reviewer for pointing out redundancy in the manuscript. We made efforts to streamline the text particularly in the Introduction section.

L55-56: Itakura et al. (2019) demonstrated that eDNA analysis allows us to reveal the spatial distribution, abundance, and biomass of Japanese eels at the river-basin scale.

(Response)

We are aware of the study of Itakura et al. (2019) as referred in the next paragraph of the former manuscript. Although their study surveying in more than 100 sites is impressive, the study not necesarily covered nation wide river basins. Besides, the topic in this first paragraph is the distribution of Japanese eel in general without a focus on eDNA. To avoid confusion we revised from “distribution” to “spatiotemporal distributions” and from “in rivers“ to “in various river sites“ in this sentence (Line 56-57).

L73-75: Is this true? I think, the relationship between eDNA concentration and biomass and/or the number of individuals is a major topic that has been examined by a great number of studies, and some of which have reported a strong relationship.

(Response)

Yes. It is true. Some researchers have reported a strong relationship between eDNA concentration and biomass and/or the number of individuals. However, other researchers, including Itakura et al. (2019), have reported only weak positive relationship between them (Line 82-83). Considering that non-significant results would be published less, we can say that the relationship between eDNA concentration and fish abundance is yet to be established.

L79: “The eDNA concentration increases 10–200× during spawning [35].” → "[25]" ?

(Response)

We thank the reviewer for pointing out this flaw. We revised the citation (Line 107).

L90: Ghosal et al. (2018) reported that the carp eDNA concentration increased 500× at night when the fish biomass only doubled. However, this experiment was carried out under special experimental conditions. Thus, descriptions with generality should be avoided.

(Response)

We agree with the reviewer to the point that Ghosal et al (2018) described a special case. Therefore, for clarity, we revised “The carp eDNA concentration increased 500× at night when the fish biomass only doubled [40]” as “Nocturnal carp eDNA concentrations increased 500-fold at night when fish biomass only doubled at a feeding site [36]” (Line 95-96).

L157, 169: “-20 °C” → “−20 °C”

(Response)

We revised them according to the suggestion (Lines 210).

L182: “Each PCR reaction system included” → “Each PCR reaction included”

(Response)

As per the reviewer’s suggestion, we revised “Each PCR reaction system included” as “Each PCR reaction included” (Line 236).

L190: “There were three replicates per sample” → There were three replicates per sample “and standard DNAs”?

(Response)

As per the reviewer’s suggestion, we added “and standard DNAs” after “There were three replicates each of all samples” (Lines 243-244).

L198: “each target species” → each target individual ?

(Response)

As per the reviewer’s suggestion, we revised “each target species” as “each target individual” (Line 252).

L174-176, L423-426: I still have a question as to why we did not use other species-specific primers and probe sets designed in other regions. Again, the specificity is not so important in this study because tank water was used in all experiments.

In the responses to the reviewers, the authors accepted that D-loop is more likely to accumulate intraspecific genetic diversity than other regions. The other mitochondrial regions can also be polymorphed, but the "mutation rate" vary widely among regions. I agree that the presence of polymorphisms is an issue to be generally careful. However, in the eDNA study, the risk of underestimation caused by the primer/probe mismatch can be reduced by selecting regions that generally have lower mutation rates than D-loop.

In my opinion, in this manuscript, the authors should include the discussion about the selection of the target region to reduce the risk of mismatches and the reason why the primer-probe set designed in the D-loop was used in this study.

(Response)

We thank the reviewer for this constructive comments. We added the discussion about the selection of the target region (D-loop) in the discussion section (Line 512-519).

This primers/probe set has been also used in Kasai et al. (2020) and Kasai. et al (2021), who conducted a nationwide eel distribution survey in Japan. Individual tank experiment of our study using this primers/probe sets revealed that polymorphism in the probe region strongly affected eDNA detection. We carefully checked the specificity in the set of primers, then we used this set in our study. It is necessary to design primers/probe set selecting lower intraspecific polymorphic regions, when eDNA concentration of each individuals must be precisely measured.

This point has been emphasized in the revised manuscript.

L394-396: The eDNA concentration was different among samples. Thus, the comparison of Ct values here is meaningless.

(Response)

As per the reviewer’s suggestion, we delated the sentence describing Ct values (Line 465-466).

・In my opinion, in all tables and Figures, eDNA concentration should be shown per volume of filtered water (500 mL) or per PCR template (2 µL).

(Response)

We consider this suggestion of the reviewer very carefully. As an independent experiment it is indeed sensible to express eDNA concentration per filtered water (500 mL) or PCR template (2 µL). Meanwhile it seems to be common to express eDNA concentraiton in copies / L (Maruyama et al. 2014; Klobucar et al. 2017; Horiuchi et al. 2019; Yates et al. 2020 etc). Furthermore, the present study is strongly related to Kasai et al. (2021) in which the same primers-probe set was used to detected eel eDNA in natural waters of 50–600 mL per site then it was converted to be copies / L. Therefore we would like to be consistent to this expression.

・Please check the following points: (1) Are coloured letters allowed in PLOS One? (2) Are the legends inserted in the correct place in the manuscript?

(Response)

We thank the reviewer for pointing them out. (1) We revised coloured letters (Line 356, 361-363, 377-379, 404). (2) Submission Guidelines of PLOS ONE says “The (figure) caption may also include a legend as needed”, a newline before the legend was added in the figure caption example, and “Place legends, footnotes, and other text below the table”. It also says “You may also include a legend in your caption, but it is not required”. Therefore, we added newlines before the legends in the figures, and delated legends from tables and figures in the Supporting Information section (Line 808-848).

Table1: Please add the Tm of the probes that are expected in each situation (Non variant, Vatiant-1, and Variant-2).

(Response)

As we use MGB (minor groove binder) probe, it is not possible to calculate the expected Tm including the effect of mismatch nucleotides between the variants and the probe sequence due to the non-disclosure of calculation mechanism (the patent is held by ThermoFisher).

Fig2-1(D): What does the Y-axis indicate?

(Response)

We thank the reviewer for pointing out this flaw. Y-axis had been hiding behind. We brought it to the front.

Fig2-1 and 2: To compare the eDNA concentration for each sample quantified by TaqMan (Fig. 2-1) and SYBR (Fig. 2-2), I recommend using sample-specific marks.

(Response)

As per the reviewer’s suggestion, we revised the figures to have sample-specific marks; square: A-1; diamond: B-1; triangle: C-1; cross: A-2; bar: B-2; circle: C-2. Figure captions have also been revised accordingly (Fig2-1 and Fig2-2).

Fig.2-1(A), 2(A): The red and orange dots are difficult to distinguish, so I recommend changing the colour.

(Response)

As per the reviewer’s suggestion, we revised the colour of orange as green in Fig.2-1(A), Fig.2-2(A).

S2 table: “between 5'-primer F and probe” → “between 3'-primer F and probe” ?

“between probe and primer R-3'” → “between probe and primer R-5'” ?

“*Green: base different from sample 0525_A-1.” → What does 0525 mean?

(Response)

We revised “between 5'-primer F and probe” as “between primer F-3' and probe”, and “between probe and primer R-3'” as ”between probe and 3'-primer R”. “0525” means sampling day.

7. PLOS authors have the option to publish the peer review history of their article (what does this mean?). If published, this will include your full peer review and any attached files.

Do you want your identity to be public for this peer review? For information about this choice, including consent withdrawal, please see our Privacy Policy.

Reviewer #1: No

---

## [Editor Report · Decision Letter 2]

21 Jul 2021

Intraspecific genetic variability and diurnal activity affect environmental DNA detection in Japanese eel

PONE-D-20-38269R2

Dear Dr. Takahashi,

We’re pleased to inform you that your manuscript has been judged scientifically suitable for publication and will be formally accepted for publication once it meets all outstanding technical requirements.

Kind regards,

Hideyuki Doi

Academic Editor

PLOS ONE

Additional Editor Comments (optional):

I carefully checked the revised manuscript as well as the response letter. I agree the revisions according to the reviewers’ comments and now can recommend to publish the paper in this journal.
---

## [Editor Report · Acceptance letter]

8 Sep 2021

PONE-D-20-38269R2 

Intraspecific genetic variability and diurnal activity affect environmental DNA detection in Japanese eel 

Dear Dr. Takahashi:

I'm pleased to inform you that your manuscript has been deemed suitable for publication in PLOS ONE. Congratulations! Your manuscript is now with our production department. 

Kind regards, 

on behalf of

Dr. Hideyuki Doi 

Academic Editor

PLOS ONE